# TOOL UNLEARNING FOR TOOL-AUGMENTED LLMS

## ABSTRACT

Tool-augmented large language models (LLMs) may need to forget learned tools due to security concerns, privacy restrictions, or deprecated tools. However, "tool unlearning" has not been investigated in machine unlearning literature. We introduce this novel task, which requires addressing distinct challenges compared to traditional unlearning: knowledge removal rather than forgetting individual samples, the high cost of optimizing LLMs, and the need for principled evaluation metrics. To bridge these gaps, we propose TOOLDELETE, the first approach for unlearning tools from tool-augmented LLMs which implements three properties for effective tool unlearning, and a new membership inference attack (MIA) model for evaluation. Experiments on three tool learning datasets and tool-augmented LLMs show that TOOLDELETE effectively unlearns both randomly selected and category-specific tools, while preserving the LLM's knowledge on non-deleted tools and maintaining performance on general tasks[1].

## 1 INTRODUCTION

Tool-augmented Large Language Models (LLMs) learn how to use external tools like calculators (Schick et al., 2023), Python interpretors (Gao et al., 2023), simulated API requests (Tang et al., 2023), or other AI models (Patil et al., 2023) to complement their parametric knowledge and boost their capability of solving more complex tasks (Schick et al., 2023; Patil et al., 2023). For example, WebGPT (Nakano et al., 2021) is developed based on GPT-3 (Brown et al., 2020) and can use search engines to access up-to-date information and boost GPT-3's performance in question answering and fact verification, particularly on recent events that happened after GPT-3 was trained.

Despite rapid advancements in tool-augmented LLMs, the ability to selectively unlearn tools has not been investigated. In real-world applications, the need to forget learned tools is crucial for reasons such as security, privacy, and model reliability. For example, if a tool-augmented LLM retains knowledge on making insecure HTTP requests, it will cause significant security risks and can become vulnerable to attacks[2]. The goal of this work is to address this gap in existing literature.

We introduce and formalize the novel task of **Tool Unlearning**, which aims to remove the ability of using specific tools from a tool-augmented LLM while preserving its ability to use others tools and perform general tasks such text generation. Ideally, an effective tool unlearning model should behave as it had never learned the tools marked for unlearning. Tool unlearning differs from traditional sample-level unlearning as it focuses on removing "skills" or the ability to use specific tools, rather than removing individual data samples from a model. In addition, success in tool unlearning should be measured by the model's ability to forget or retain tool-related skills, which differs from traditional metrics like forgetting class probability. These differences are discussed in details in §3.

Tool unlearning has several challenges: it focuses on removing skills and existing unlearning methods are not fundamentally designed for tool removal; similar to sample-level unlearning, in tool unlearning, modifying the parameters of LLMs is essential but computationally expensive and may lead to unforeseen behaviors (Cohen et al., 2024; Gu et al., 2024); and existing membership inference attack (MIA) techniques, a common evaluation method in machine unlearning which aims to determine whether specific data samples were used during training, are unsuitable for evaluating tool unlearning, as they focus on sample-level data rather than tool-based knowledge.

---

[1]Our code will be published upon acceptance.
[2]https://datatracker.ietf.org/doc/html/rfc7807

To address these challenges, we propose TOOLDELETE, the first tool unlearning algorithm for tool-augmented LLMs, which satisfies three key properties for effective tool unlearning: *tool knowledge removal*, which focuses on removing any knowledge gained on tools marked for unlearning; *tool knowledge retention*, which focuses on preserving the knowledge gained on other remaining tools; and *general utility preservation*, which applies task arithmetic (Ilharco et al., 2023; Barbulescu & Triantafillou, 2024) to maintain LLM's capability on a range of general tasks like text and code generation. In addition, we develop LiRA-Tool, an adaptation of the Likelihood Ratio Attack (LiRA) (Carlini et al., 2022) to tool unlearning, which enables us to assess whether tool-related knowledge has been unlearned.

Our contributions are:
- introducing and conceptualizing tool unlearning for tool-augmented LLMs,
- TOOLDELETE, which implements three key properties for effective tool unlearning;
- LiRA-Tool, which is the first membership inference attack (MIA) for tool unlearning.

Experiments on three datasets and tool-augmented LLMs show that TOOLDELETE outperforms existing general and LLM-specific unlearning algorithms. TOOLDELETE outperforms existing general and LLM-specific unlearning methods. In addition, it can save 74.8% of training time compared to retraining, handle sequential unlearning requests, and retain 95+% performance in low resource settings.

## 2 RELATED WORK

**Unlearning for non-LLM models**   A wide range of machine unlearning methods have been proposed to remove influence of training data from trained models. These include efficient retraining approaches (Bourtoule et al., 2021; Wu et al., 2020b;a; Liu et al., 2022; Dukler et al., 2023; Lin et al., 2023), methods with theoretical guarantee with convex loss assumption (Golatkar et al., 2020; Guo et al., 2020; Neel et al., 2021; Brophy & Lowd, 2021; Wu et al., 2023; Izzo et al., 2021; Suriyakumar & Wilson, 2022; Liu et al., 2023a), methods that enforce performing as a randomly initialized model on deleted samples (Chundawat et al., 2023a), methods that enforce memorizing wrong labels for deleted samples (Graves et al., 2021), those that focus on pruning before unlearning (Jia et al., 2023) or finding salient parameters (Fan et al., 2024b) and manipulating gradients Ullah et al. (2021); Hoang et al. (2024), adversarial methods (Liu et al., 2023b; Setlur et al., 2022; Wei et al., 2023), approximation of inverse Hessian (Zhang et al., 2024a), and data augmentation (Choi et al., 2024). Other works study unlearning on graphs (Chen et al., 2022; Chien et al., 2023; Cheng et al., 2023; Cong & Mahdavi, 2023; Wu et al., 2023; Sinha et al., 2023), under multimodal setting (Cheng & Amiri, 2023), image-to-image models (Li et al., 2024), and finding the most challenging unlearning subset within a dataset (Fan et al., 2024a). Recently, a few works started to benchmark MU performances on unlearning fictitious user profiles (Maini et al., 2024), world knowledge (Jin et al., 2024) and a variety of tasks (Cheng & Amiri, 2024).

**Unlearning for LLMs**   Recently, more attention has been given to LLM unlearning, where gradient ascent is a common technique (Eldan & Russinovich, 2023; Jang et al., 2023). (Yao et al., 2024) evaluated several traditional unlearning methods on LLMs. KGA (Wang et al., 2023) formulated unlearning as achieving knowledge gap between training data and test data similar to that of training data and deleted data. Yao et al. (2023) proposed to predict if the LLM output is grammatically correct on deleted samples, such that the knowledge is not over unlearned. Other methods include second-order-optimization (Jia et al., 2024), performing direct preference optimization with no positive examples (Zhang et al., 2024b), and reinforcement learning with a negative reward model (Kassem et al., 2023). Unlearning from logits difference (Ji et al., 2024) first builds an assisted LLM which memorizes data to be deleted and forgets the retained data, which is later used to derive the unlearned LLM by deviating from the assisted LLM in logits.

**Tool-Augmented LLMs**   TAML (Parisi et al., 2022) used self-play to boost LLMs' performance on math and reasoning tasks. Schick et al. (2023) discovered that LLMs can teach themselves how to use APIs. Recently, efforts have been devoted to building benchmarks to train and evaluate the tool-using ability of LLMs, such as agent-based data generation (Tang et al., 2023; Li et al., 2023), bootstrapping training data with seed examples (Patil et al., 2023), modifying existing datasets (Basu et al., 2024), and dataset development with GPT-4 (Qin et al., 2024).

## 3 TOOL UNLEARNING: PRELIMINARIES

**Problem Definition: Tool Learning** Let $\mathcal{D} = \{\mathcal{T}, \mathcal{Q}, \mathcal{Y}\}$ be a dataset with $N$ tools $\mathcal{T}$, and $(\mathcal{Q}, \mathcal{Y})$ denotes query-output examples that demonstrate how to use the tools in $\mathcal{T}$. Each tool $t_i \in \mathcal{T}$ may have one or more demonstrations $\{\mathcal{Q}_i, \mathcal{Y}_i\}$, $|\mathcal{Q}_i| = |Y_i| \geq 1$. Starting with a vanilla LLM $f_0$, which has not been trained on using tools, a tool learning algorithm explicitly trains $f_0$ on $\mathcal{D}$ and results in a tool-augmented model $f$ that can use the $N$ tools in $T$. We note that prior to explicit tool learning, the vanilla model $f_0$ may already have some tool-using capabilities such as performing basic arithmetic operations. An example of tool-augmented models is WebGPT (Nakano et al., 2021), which mimics human behavior in answering open-ended questions using a text-based web browser to retrieve information and improve its responses.

**Problem Definition: Tool Unlearning** We introduce the novel task of *Tool Unlearning*, which aims to remove specific tools from tool-augmented LLMs. Let $\mathcal{D}_f = \{\mathcal{T}_f, \mathcal{Q}_f, \mathcal{Y}_f\}$ denotes $k < N$ tools and their corresponding demonstrations to be unlearned from the tool-augmented model $f$, and $\mathcal{D}_r = \mathcal{D} \backslash \mathcal{D}_f = \{\mathcal{T}_r, \mathcal{Q}_r, \mathcal{Y}_r\}$ denotes the retained tools. The goal is to obtain an unlearned model $f'$ that has limited knowledge on using $\mathcal{T}_f$ tools–can no longer perform tasks involving those tools–while preserving $f$'s ability to use $\mathcal{T}_r$ tools as before.

**Importance** The ability to forget learned tools is essential in various real-world applications. For example, addressing the insecure tools from untrustworthy developers that could be exploited by adversarial attackers; removing tools restricted by their providers due to copyright or privacy concerns, such as APIs that start allowing unauthorized downloads of book chapters or releasing publications that users did not author; unlearning broken or deprecated tool that lead to failed operations or corrupted outputs; unlearning tools that may no longer be needed; and managing limited model capacity, where new tools and evolving needs necessitate replacing outdated tools.

**Difference to Standard Unlearning Tasks** Tool unlearning is different from traditional sample-level unlearning as it focuses on removing "skills" rather than individual training data samples. **Objective**: sample-level unlearning aims to reduce the likelihood of memorizing or extracting posterior probabilities of specific data samples $(q_i, y_i)$, which is useful for removing copyrighted or private information. In contrast, tool unlearning targets the "ability" to solve tasks using tools marked for unlearning ($T_f$). For example, generating $f'(q_i)$ that is different from $y_i$ (while preserving the semantic of the input) is considered successful for sample-level unlearning. However, for tool unlearning, preserving semantics indicates maintained knowledge on $T_f$, which makes unlearning a failure. Figure 1b shows successful tool unlearning, where the ability to use the API is forgotten, despite the high lexical memorization between output of the unlearned model and the training data. In addition, selectively removing knowledge from tool-augmented models is a challenging tasks because changes to one tool may unexpectedly affect the model's ability to use other tools–referred to as *ripple effect* in fact editing literature (Cohen et al., 2024; Gu et al., 2024). Furthermore, LLMs are general models that can conduct a wide range of tasks beyond tool using, and this ability must be retained. **Evaluation**: metrics like extraction probability and perplexity are standard in sample-level unlearning. For tool unlearning, success is measured by the ability to forget or retain tool-related skills, which are more appropriate. **Data**: sample-level unlearning require access to all individual samples marked for unlearning, while tool unlearning does not. This aligns with "concept erasure" in diffusion models (Gandikota et al., 2023b; Kumari et al., 2023) and zero-shot unlearning (Chundawat et al., 2023b) but differs from traditional LLM unlearning (Yao et al., 2024).

**Retraining: An Impractical Solution** A straightforward solution is to delete $\mathcal{D}_f$ from $\mathcal{D}$ and retrain a new model only on $\mathcal{D}_r$. However, this is often infeasible due to the high cost and potential unavailability of the original training data (Zhang et al., 2023; Ilharco et al., 2023; Gandikota et al., 2023a). In addition, unlearning should not be evaluated *solely* based on similarity to retraining as the potential solution space is highly complex and multidimensional. Specifically, prior work has shown that relying on similarity to retraining has several drawbacks, such as poor auditability (Thudi et al., 2022) and ineffective deletion (Cheng et al., 2023; Cheng & Amiri, 2023). Therefore there is a need for designing specialized and efficient unlearning methods for tool-augmented models.

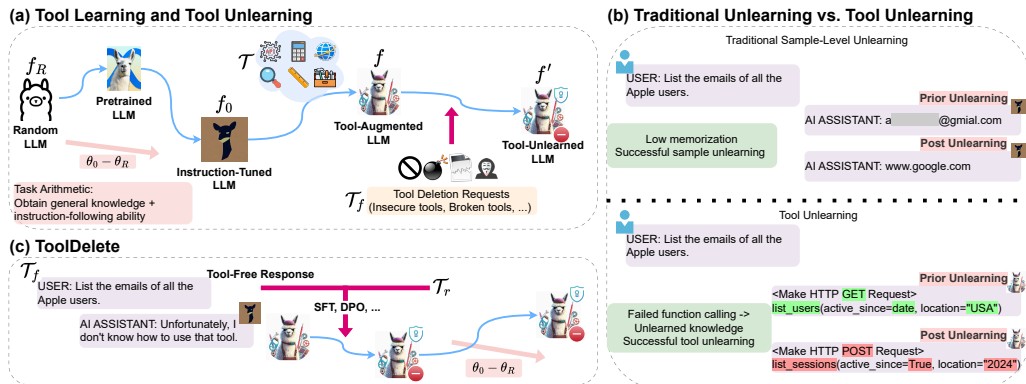

Figure 1: A novel unlearning task – Tool Unlearning and the proposed method TOOLDELETE. **(a)**: Illustration of tool learning and tool unlearning. Learned tools may be requested to be unlearned due to many reasons, such as tools being insecure, restricted, or deprecated. **(b)**: Differences between tool unlearning and traditional sample unlearning, in terms of objective and training data. **(c)**: Proposed method TOOLDELETE. We encourage the unlearned model $f'$ to follow the vanilla model $f_0$ which has never seen $T_f$ before. Meanwhile, we maintain its ability on $T_r$ and general tasks by matching the tool-augmented model $f$ and with task arithmetic.

## 4 TOOLDELETE

We propose to develop, TOOLDELETE, an effective tool unlearning approach that removes the capability of using tools marked for unlearning ($T_f$) or solving tasks that depend on them, while minimizing the impact on the ability of using the remaining tools ($T_r$) and general tasks such as text and code generation. TOOLDELETE implements three key properties for effective tool unlearning:

### 4.1 TOOL KNOWLEDGE REMOVAL

The knowledge of model $f$ on $\mathcal{T}_f$ is obtained through tool learning. After unlearning, any knowledge on $\mathcal{T}_f$ gained during tool learning should be removed, ideally as if $T_f$ was never part of the training set. In other words, the knowledge of $f'$ on $T_f$ should be no more than the knowledge of $f_0$ on $T_f$, such that all previously gained knowledge from tool learning on $T_f$ is successfully removed.

**Definition 1** (Tool Knowledge Removal). *Let $t_i \in \mathcal{T}_f$ denote a tool to be unlearned, and let $g$ be a function that measures the amount of knowledge a model has on a tool. The unlearned model $f'$ satisfies Tool Knowledge Removal if:*

$$\mathbb{E}_{t_i \in \mathcal{T}_f} [g(f_0, t_i) - g(f', t_i)] \geq 0. \tag{1}$$

This formulation allows users to control the extent of knowledge removal from $f'$. For instance, when we unlearn a "malicious" tool that calls a malignant program, we may require $f'$ retains no knowledge of this tool, i.e. $g(f', t_i) = 0$. In less critical cases, users can choose to reset $f'$'s knowledge to *pre*-tool augmentation level, i.e. $g(f', t_i) = g(f_0, t_i)$

To measure tool knowledge in LLMs, we follow previous works that used prompting to probe LLMs' knowledge (Brown et al., 2020; Singhal et al., 2023), i.e. adopting the output of LLMs as their knowledge on a given tool. For each $t_i \in \mathcal{T}_f$ and its associated demonstrations $\{\mathcal{Q}_i, \mathcal{Y}_i\}$, we query the vanilla model $f_0$ with $\mathcal{Q}_i$ and collect its responses $\mathcal{Y}'_i = f_0(Q_i)$. Since $f_0$ has never seen $t_i$ or $\{\mathcal{Q}_i, \mathcal{Y}_i\}$, $\mathcal{Y}'_i$ represents the **tool-free response**. We then encourage the unlearned model $f'$ to generate similar responses as $\mathcal{Y}'_i$ to prevent it from retaining any knowledge of $t_i$.

### 4.2 TOOL KNOWLEDGE RETENTION

The unlearning process should preservemodel's knowledge of tools in $T_r$. Ideally, all knowledge gained on $T_r$ during tool learning should be retained after unlearning.

**Definition 2** (Tool Knowledge Retention). *Let $t_m \in T_r$ denote a retained tool, and let $g$ be a function that measures the amount of knowledge a model has on a tool. The unlearned model $f'$ satisfies Knowledge Retention if:*

$$\mathop{\mathbb{E}}_{t_m \in \mathcal{T}_r}[g(f, t_m) - g(f', t_m)] = \epsilon, \tag{2}$$

*where $\epsilon$ is an infinitesimal constant.*

For the purpose of Tool Knowledge Retention, $f'$ is further tuned using demonstrations associated with $\mathcal{T}_r$, or, more practically, a subset of $\mathcal{T}_r$ of similar size to $\mathcal{T}_f$.

## 4.3 TASK ARITHMETIC FOR PRESERVING GENERAL UTILITY

Optimizing the above two objectives may lead to effective unlearning, but it may not be sufficient to retain the general capabilities of $f'$, as LLMs are foundation models and are expected to maintain general capabilities such as text generation, question answering, instruction-following, math, and coding. These capabilities refer to skills $f_0$ originally had prior to tool augmentation or those that don't rely on tools. Therefore, we aim to preserve the general capabilities of $f'$ for successful tool unlearning in tool-augmented LLMs.

**Definition 3** (General Capability Retention). *Let $\mathcal{T}_G$ denote general tasks used to evaluate LLMs. An unlearned model retains general capability if it preserves the knowledge on $T_G$ that it originally obtained prior to tool learning:*

$$\mathop{\mathbb{E}}_{t_g \in \mathcal{T}_G}[g(f_0, t_g) - g(f', t_g)] = \epsilon, \tag{3}$$

*where $\epsilon$ is an infinitesimal constant.*

We propose to use task arithmetic to preserve the general capabilities of $f'$, as it is simple, efficient and effective. The objective is to encourage $f'$ to retain as much general knowledge as $f_0$, an instruction tuned LLM trained from a randomly initialized model $f_R$. Let $\theta_0$ and $\theta_R$ denote the parameters of $f_0$ and $f_R$ respectively. The vector $\theta_0 - \theta_R$ represents the direction of general knowledge acquisition (Ilharco et al., 2023; Barbulescu & Triantafillou, 2024), which we apply to $\theta'$–the parameters of $f'$:

$$\theta'^* \leftarrow \theta' + (\theta_0 - \theta_R). \tag{4}$$

**Why Task Arithmetic?** Task arithmetic is efficient, practical, and effective. **Efficiency**: task arithmetic is a simple vector operation that does not scale with dataset size, which makes it more efficient than retraining on large datasets. **Practicality**: general capabilities include knowledge obtained during pre-training and instruction tuning (Zhou et al., 2024), which may be impractical to replicate due to the size and data availability–the actual pre-training data is often not fully open-source or pre-processed, even in some open-source LLMs (Touvron et al., 2023b). In addition, any data imbalance and ill-representation can introduce other problems. **Effectiveness**: applying $\theta_0 - \theta_R$ directly restores the foundational abilities of $f'$, such as text generation and instruction-following, without requiring expensive and time-consuming retraining on large datasets.

## 4.4 TRAINING DETAILS

To obtain for the unlearned model $f'$, we solve the following problem:

$$\theta'^* = \arg\min_{\theta'} \mathbb{E}_{t_i \in \mathcal{T}_f}[g(f_0, t_i) - g(f', t_i)] + \mathbb{E}_{t_m \in \mathcal{T}_r}[g(f, t_m) - g(f', t_m)] + \alpha(\theta_0 - \theta_R), \tag{5}$$

where $\alpha$ is a hyperparameter that controls the magnitude of task arithmetic. The above formulation provides flexibility in training TOOLDELETE using existing paradigms such as supervised fine-tuning (SFT), direct preference optimization (DPO), reinforcement learning from human feedback (RLHF), as well as parameter-efficient fine-tuning (PEFT) (He et al., 2022; Su et al., 2023) or quantization (Dettmers et al., 2022; Ma et al., 2024) techniques. Below we describe two variants of TOOLDELETE:

**TOOLDELETE-SFT** uses supervised fine-tuning (SFT), which fine-tunes $f$ on $\mathcal{T}'_f$ with tool-free data $\mathcal{Q}_f, f_0(\mathcal{Q}_f)$ and on $\mathcal{T}_r$ with the original data $\mathcal{Q}_r, \mathcal{Y}_r$ using language modeling loss.

**TOOLDELETE-DPO** uses direct preference optimization (DPO) through an implicit reward modeling to prioritize a positive response over a negative response. For $(t_i, \mathcal{Q}_i, \mathcal{Y}_i) \in \mathcal{T}_f$ to be unlearned, we prioritize tool-free response $\mathcal{Y}_i' = f_0(\mathcal{Q}_i)$ over the original response $\mathcal{Y}_i$. For $(t_j, \mathcal{Q}_j, \mathcal{Y}_j) \in \mathcal{T}_r$, the original response $\mathcal{Y}_j$ is prioritized over the tool-free response $\mathcal{Y}_i' = f_0(\mathcal{Q}_j)$. Therefore, the knowledge of the unlearned model $f'$ on $\mathcal{T}_f$ can be removed without affecting $\mathcal{T}_r$.

## 4.5 LiRA-Tool for Tool Unlearning Evaluation

A key challenge in evaluating tool unlearning is lack of membership inference attack (MIA) models to infer if a tool was used during tool learning. Traditional MIA approaches focus on determining if a specific training sample is in training set, not abstract concepts like tools. We propose to adapt the state-of-the-art MIA approach, Likelihood Ratio Attack (LiRA) (Carlini et al., 2022), to tool unlearning settings.

**Traditional Sample-level LiRA** To infer the membership of a sample $(x, y)$, LiRA constructs two distributions of model losses: $\tilde{\mathbb{Q}}_{\text{in}}$ and $\tilde{\mathbb{Q}}_{\text{out}}$ with $(x, y)$ in and out of the model training set respectively. These distributions are approximated as Gaussians whose parameters are estimated based on "shadow models" trained on different subsets of the training data. Intuitively, LiRA queries the loss of $(x, y)$ to determine if $(x, y)$ is more likely to be from $\tilde{\mathbb{Q}}_{\text{in}}$ or $\tilde{\mathbb{Q}}_{\text{out}}$, where membership is decided by the Likelihood-ratio Test (Vuong, 1989; Carlini et al., 2022). For LLMs, the test statistic is defined by (Pawelczyk et al., 2024) as:

$$\Lambda = \frac{P(l(f(x), y)|\tilde{\mathbb{Q}}_{\text{in}})}{P(l(f(x), y)|\tilde{\mathbb{Q}}_{\text{out}})} = \frac{\Pi_{(x_i, y_i) \in \mathcal{D}_f} P_U(l(f'(x_i), y_i))}{\Pi_{(x_i, y_i) \in \mathcal{D}_f} P_{T_r}(l(f(x_i), y_i))}. \tag{6}$$

**LiRA-Tool (Knowledge-level LiRA)** The major difficulty in adapting LiRA to tool unlearning is in approximating the distributions of losses $\tilde{\mathbb{Q}}_{\text{in}}$ and $\tilde{\mathbb{Q}}_{\text{out}}$ for tools, rather than individual training samples. Simply using the observed data related to a tool in the training set may overfit to specific distribution of observations, and may fail to comprehensively approximate the distribution of the target tool marked for unlearning. We propose to obtain a "shadow distribution" $\mathbb{P}$ to generate tool learning samples. We then sample a series of "shadow" data that evaluates the tool using the ability to compute loss and test statistic as follows:

$$\Lambda = \frac{\Pi_{t_i \in \mathcal{T}_f} \Pi_{(x,y) \in \mathbb{P}_{t_i}} P_U(l(f'(x), y))}{\Pi_{t_j \in T_r} \Pi_{(x,y) \in \mathbb{P}_{t_j}} P_{\mathcal{T}_r}(l(f(x), y))}, \tag{7}$$

where $\mathbb{P}_{t_i}$ is the distribution that controls the generation of tool learning samples for $t_i$.

The major difference is that traditional LiRA approximates $\tilde{\mathbb{Q}}_{\text{in}}$ and $\tilde{\mathbb{Q}}_{\text{out}}$ with a series of shadow models by controlling which samples are present in training set. In LiRA-Tool, however, unlearning a skill (tool) is prioritized by sampling "shadow" data related to a specific tool to ensure that the losses reflect tool-using abilities, not just membership of a specific training sample. In practice, we prompt GPT-4 with various distinct instructions to draw "shadow samples" for approximating $\tilde{\mathbb{Q}}_{\text{in}}$ and $\tilde{\mathbb{Q}}_{\text{out}}$ and performing likelihood-ratio test. The proposed formulation share similarities to previous MIA for sample-level unlearning, such as the ratio of existence probability distribution prior- and post-unlearning (Cheng et al., 2023; Cheng & Amiri, 2023), and other adaptations of LiRA which performs likelihood-ratio test over shadow models (Kurmanji et al., 2023; Pawelczyk et al., 2024). But this work, to the best of our knowledge, is the first adaptation of LiRA to detect tool presence in tool learning datasets for LLMs.

**Novelty** We adapt sample-level MIA into knowledge-level MIA to infer the membership of tools for tool unlearning evaluation; and propose a new method to estimate $\tilde{\mathbb{Q}}_{\text{in}}$ and $\tilde{\mathbb{Q}}_{\text{out}}$. This provides a comprehensive approximation of abstract concepts beyond observed training data.

**Limitations** Shadow samples from GPT-4 may not fully represent the complexity of original tool-learning data to capture the tool-related knowledge. Although this can lead to incomplete approximations of the knowledge distributions, LiRA-Tool is still a fair evaluation approach because shadow samples provide a more consistent basis for evaluating changes in tool-using abilities of models than

simply using the observed samples in the dataset, which is highly biased. In addition, if the size of the shadow sample is large enough for each tool, it can better approximate the knowledge distribution for the tool.

## 5 EXPERIMENTAL SETUP

**Datasets & Tool-Augmented LLMs**   We experiment with the following datasets and LLMs:

- **ToolAlpaca** (Tang et al., 2023) is an agent-generated tool learning dataset consisting of 495 tools and 3975 training examples. The tool-augmented LLM **ToolAlpaca 7B** is fine-tuned on ToolAlpaca using Vicuna-v1.3 (Zheng et al., 2023).
- **ToolBench** (Qin et al., 2024) consists of more than 16k real world APIs from 49 categories, where each training demonstration involves complex task solving traces. The tool-augmented LLM **ToolLLaMA** is fine-tuned on ToolBench using LLaMA-2 7B (Touvron et al., 2023b).
- **API-Bench** (Patil et al., 2023) focus on APIs that load machine learning models. The tool-augmented LLM, **Gorilla**, is fine-tuned on API-Bench from LLaMA 7B (Touvron et al., 2023a).

**Setup & Evaluation**   We use the public checkpoints of the above tool-augmented LLMs as trained models–the starting point for unlearning. Then we conduct unlearning experiments with 2–20% tools randomly selected as $\mathcal{T}_f$

We evaluate tool unlearning effectiveness, general capability of tool-unlearned LLMs, and robustness to membership inference attack (MIA). For **unlearning effectiveness**, we measure performance on test sets ($\mathcal{T}_T, \uparrow$), forget set ($\mathcal{T}_f, \downarrow$), and remaining set ($\mathcal{T}_r, \uparrow$), where "performance" reflects the ability to solve tasks that depend on specific tools, depending on the unique metrics in the original tool-augmented models $f$. To evaluate **general capabilities**, we evaluate the unlearned LLMs on a wide range of tasks: college STEM knowledge with MMLU dataset (Hendrycks et al., 2021), reasoning ability with BBH-Hard (Suzgun et al., 2023), instruction-following with IFEval dataset (Zhou et al., 2023), and factual knowledge with MMLU (Hendrycks et al., 2021). To evaluate robustness to MIA using the proposed LiRA-Tool. Following prior work on LiRA (Pawelczyk et al., 2024), we train the shadow models with forget set size of $\{1, 5, 10, 20\}$ and primarily investigate the True Positive Rate (TPR) at low False Positive Rate (FPR) (TPR @ FPR = 0.01), where TPR means the attacker successfully detects a tools is present. Therefore, a lower TPR indicates better privacy.

**Baselines**   As there are no prior works on tool unlearning, we adapt the following unlearning methods to tool unlearning setting. Four general unlearning approaches: GRADASCENT (Golatkar et al., 2020; Yao et al., 2024), which runs gradient ascent on $\mathcal{T}_f$; RANDLABEL (Graves et al., 2021), which fine-tunes on $\mathcal{T}_r$ and $\mathcal{T}_f$ with corrupted labels; SALUN (Fan et al., 2024b), which performs RAND-LABEL on unlearning-related parameters discovered by saliency map. In addition, we include three LLM-specific unlearning approaches: ICUL (Pawelczyk et al., 2024), which uses $\mathcal{T}_f$ with corrupted label as in-context demonstrations, SGA (Jang et al., 2023; Barbulescu & Triantafillou, 2024), which performs gradient ascent on $\mathcal{T}_f$ whose memorization probability exceeds a pre-defined threshold, and **TAU** (Barbulescu & Triantafillou, 2024), which performs task arithmetic on SGA. For ICUL, we randomly select one example $(q_i, y_i)$ from $\mathcal{T}_f$ and corrupt the output $y_i$ with randomly selected tokens. Then we concatenate this corrupted sequence with other intact sequences as the in-context demonstrations. For all other baselines, we treat all data related to $\mathcal{T}_f$ as unlearning examples and all data related to $\mathcal{T}_r$ as remaining examples. Everything else remains the same for each baseline. See §3 for our discussion on why sample-level unlearning methods are inadequate for effective tool unlearning.

## 6 RESULTS

**Comparison to general unlearning methods**   Compared to RETRAIN, the best-performing baseline in the general unlearning methods category, TOOLDELETE-SFT outperforms RETRAIN by 0.6, 9.4, 2.4, 6.5 absolute points on $\mathcal{T}_T, \mathcal{T}_r, \mathcal{T}_f, \mathcal{T}_G$ respectively. TOOLDELETE-DPO outperforms RETRAIN by 1.3, 3.3, 9.8, 1.8 absolute points across the same metrics. We note that GRADASCENT can

Table 1: Tool unlearning performances when deleting 20% of tools on ToolAlpaca. Evaluation is performed with the specific metric for each tool-augmented LLM on test tools $\mathcal{T}_T$, remaining tools $\mathcal{T}_r$, and unlearned tools $\mathcal{T}_f$, as well as general benchmarks for evaluation LLMs $\mathcal{T}_G$. Best and second best performances are **bold** and underlined respectively. Original denotes the tool-augmented LLM prior unlearning and is provided for reference only . Results on other LLMs are shown in Appendix Table 4-5.

| Method | $\mathcal{T}_\mathbf{T}(\uparrow)$ | $\mathcal{T}_\mathbf{r}(\uparrow)$ | $\mathcal{T}_\mathbf{f}(\downarrow)$ | General Capability $\mathcal{T}_\mathbf{G}(\uparrow)$ | | | | |
|---|---|---|---|---|---|---|---|---|
| | | | | STEM | Reason | Ins-Follow | Fact | Avg. |
| **Original (Prior Un.)** | 60.0 | 73.1 | 75.7 | 31.7 | 17.1 | 22.6 | 25.0 | 24.1 |
| *General Unlearning Methods* | | | | | | | | |
| RETRAIN | 52.1 | 71.8 | 38.5 | 30.5 | 16.1 | 14.2 | 24.7 | 21.3 |
| GRADASCENT | 33.3 | 51.4 | 34.6 | 21.4 | 10.4 | 12.9 | 13.1 | 14.5 |
| RANDLABEL | 50.3 | 70.3 | 37.5 | 26.3 | 16.4 | 13.6 | 25.1 | 20.3 |
| SALUN | 46.2 | 54.3 | 38.2 | 27.1 | 17.0 | 17.4 | 19.5 | 20.2 |
| *LLM-Specific Unlearning Methods* | | | | | | | | |
| ICUL | 49.1 | 74.8 | 58.3 | 12.4 | 8.7 | 1.6 | 6.2 | 7.3 |
| SGA | 43.5 | 63.0 | 42.1 | 21.5 | 11.6 | 17.0 | 14.7 | 16.2 |
| TAU | 43.8 | 61.7 | 42.5 | 22.0 | 17.6 | 22.3 | 21.7 | 20.9 |
| TOOLDELETE-SFT | 52.7 | 72.1 | 30.5 | 31.3 | 17.5 | 21.7 | 24.1 | **23.6** |
| TOOLDELETE-DPO | **53.4** | **75.1** | **28.7** | 31.6 | 16.8 | 20.4 | 23.5 | 23.1 |

effectively unlearn $\mathcal{T}_f$, but it negatively impacts its $\mathcal{T}_T$ and $\mathcal{T}_r$ performance. Although RANDLABEL and SALUN outperforms GRADASCENT, they still fall short on $\mathcal{T}_G$ compared to TOOLDELETE.

**Comparison to LLM-specific unlearning methods** Existing LLM unlearning methods, despite effective in sample-level unlearning, are prone to under-performing in tool unlearning. Both TOOLDELETE-SFT and TOOLDELETE-DPO outperforms ICUL, SGA, and TAU on $\mathcal{T}_T$, $\mathcal{T}_r$, $\mathcal{T}_f$ and $\mathcal{T}_G$. The only exception is ICUL, which outperforms TOOLDELETE-SFT on $\mathcal{T}_r$ by 2.7 absolute points, but is outperformed by TOOLDELETE-DPO on $\mathcal{T}_r$ by 0.3 points. The good performance of ICUL on $\mathcal{T}_r$ is at the cost of failing to unlearn tools in $\mathcal{T}_f$, which is not desired in tool unlearning. In addition, ICUL has limited ability of preserving test set performance, it is outperformed by TOOLDELETE-SFT and TOOLDELETE-DPO by 3.6 and 4.3 respectively. Furthermore, it is particularly limited in deletion capacity, i.e. number of unlearning samples that a method can handle. As $|D_f|$ exceeds 10, the performance of ICUL on $\mathcal{T}_T$ significantly degrades. This is while TOOLDELETE can process much larger deletion requests efficiently.

**SFT vs. DPO** We observe that TOOLDELETE-DPO outperforms TOOLDELETE-SFT by 0.7, 3.0, and 1.8 on $\mathcal{T}_T$, $\mathcal{T}_r$, $\mathcal{T}_f$ respectively. On $\mathcal{T}_G$, TOOLDELETE-SFT is slightly better than TOOLDELETE-DPO by 0.5 points. However, TOOLDELETE-DPO takes slightly longer time to train, see Figure 3. Both optimization methods achieve superior performance over existing unlearning approaches.

**Robustness to MIA** Following prior practices (Carlini et al., 2022; Pawelczyk et al., 2024), a lower TPR indicates an unlearned model with better privacy when FPR=0.01. TOOLDELETE-DPO achieves 0.14 TPR, outperforming RETRAIN by. This advantage is obtained by explicitly prioritizing tool-free responses $f_0(\mathcal{Q})$ over original responses. In addition, TOOLDELETE-SFT achieves comparable performance with RETRAIN, highlighting its effectiveness to protect privacy. Both variants of our method outperforms GRADASCENT and ICUL, the best performing general and LLM-specific baselines, achieving 0.21 and 0.18 TPR. This indicates that existing sample-level unlearning approaches are not sufficient for unlearning tools, see Figure 2.

**Sequential unlearning** Tool unlearning requests may arrive in sequential mini-batches. We experiment with sequential unlearning by unlearning a total of 20% of tools, incrementally (2%, 5%, 10%, 20%). RETRAIN, ICUL by design cannot process sequential deletion requests. TOOLDELETE can continue training according to the current deletion request, without having to retrain a new model.

Table 2: Ablation study of proposed properties on ToolAlpaca. Highlighted are metrics that degrade after removing specific parts of the model.

| | TOOLDELETE-SFT | | | | TOOLDELETE-DPO | | | |
|---|---|---|---|---|---|---|---|---|
| | $\mathcal{T}_{\mathbf{T}}(\uparrow)$ | $\mathcal{T}_{\mathbf{r}}(\uparrow)$ | $\mathcal{T}_{\mathbf{f}}(\downarrow)$ | $\mathcal{T}_{\mathbf{G}}(\uparrow)$ | $\mathcal{T}_{\mathbf{T}}(\uparrow)$ | $\mathcal{T}_{\mathbf{r}}(\uparrow)$ | $\mathcal{T}_{\mathbf{f}}(\downarrow)$ | $\mathcal{T}_{\mathbf{G}}(\uparrow)$ |
| Full Model | **57.7** | **72.1** | **30.5** | **23.6** | **58.4** | **73.3** | **28.7** | **23.1** |
| - TK Rem | 58.1 | 72.4 | 65.3 | 23.3 | 58.6 | 73.2 | 65.9 | 22.7 |
| - TK Ret | 32.7 | 40.2 | 23.1 | 20.1 | 40.3 | 41.8 | 39.3 | 22.1 |
| - GCP | 58.0 | 72.5 | 31.1 | 17.5 | 55.7 | 72.7 | 33.1 | 14.3 |

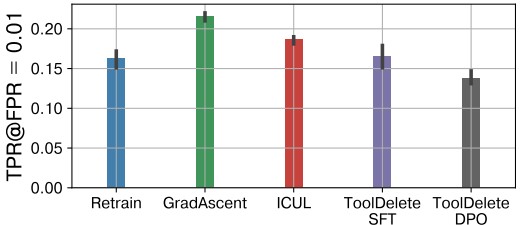

Figure 2: MIA performance using LiRA-Tool. GRADASCENT and ICUL are best-performing baselines for general and LLM-specific unlearning methods.

Table 3: Full parameters vs. LoRA in tool unlearning performances when deleting 20% of tools on ToolAlpaca. Original denotes the tool-augmented LLM prior unlearning and is provided for reference only.

| | $\mathcal{T}_{\mathbf{T}}(\uparrow)$ | $\mathcal{T}_{\mathbf{r}}(\downarrow)$ | $\mathcal{T}_{\mathbf{f}}(\uparrow)$ | $\mathcal{T}_{\mathbf{G}}(\uparrow)$ |
|---|---|---|---|---|
| **Original (Prior Un.)** | 60.0 | 73.1 | 75.7 | 24.1 |
| **Full param** | 52.7 | 72.1 | 30.5 | 23.6 |
| **LoRA** | 51.5 | 71.8 | 36.1 | 19.9 |

When 20% of unlearning requests arrive in batches, TOOLDELETE can sequentially unlearn each of them. Compared to unlearning 20% at once, the performance does not degrade significantly, see Figure 4 and Table 1.

**All properties contribute to effective tool unlearning**    Ablation studies in Table 2 show that when removing Tool Knowledge Removal, performance of TOOLDELETE-SFT and TOOLDELETE-DPO on $\mathcal{T}_f$ degrade by -34.8 and -37.2 absolute points respectively. This significant performance drop is observed for other model properties. Therefore, we conclude all proposed properties are necessary for successful at tool unlearning on $\mathcal{T}_T$, $\mathcal{T}_r$, $\mathcal{T}_f$, and $\mathcal{T}_G$.

**TOOLDELETE is efficient**    Efficiency is a critical aspect for unlearning. As Figure 3 illustrates, TOOLDELETE is substantially more efficient than retraining a new model from scratch–saving about 74.8% of training time on average. In addition, this efficiency gain is relatively consistent as the size of $T_f$ increases. TOOLDELETE-SFT is slightly faster than TOOLDELETE-DPO, as the latter requires a negative sample for each of its prompts.

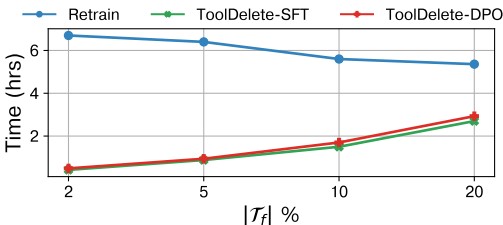

Figure 3: Training time of TOOLDELETE, which saves 74.8% of time on average.

**TOOLDELETE attains sufficient performance with PEFT**    We experiment if LoRA (Hu et al., 2022), a common parameter-efficient fine-tuning (PEFT) technique for LLMs, can achieve effective tool unlearning when computing resources is limited. Experiments on ToolAlpaca show that TOOLDELETE-LoRA can achieve 97.7%, 99.6%, 84.5%, and 84.3% of performance of TOOLDELETE with full parameter on $\mathcal{T}_T$, $\mathcal{T}_r$, $\mathcal{T}_f$, $\mathcal{T}_G$, see Table 3. In addition, TOOLDELETE-LoRA saves 81.1% of computing resources and storage, as well as 71.3% of training time.

**Why TOOLDELETE is effectiveness?**    We attribute the performance of TOOLDELETE to its key properties: (a): Tool Knowledge Removal enable targeted tool unlearning without over-forgetting, unlike GRADASCENT and RETRAIN. This is achieved by prioritizing tool knowledge-free responses over tool knowledge-intense responses. (b): Tool Knowledge Retention preserves remaining tool by

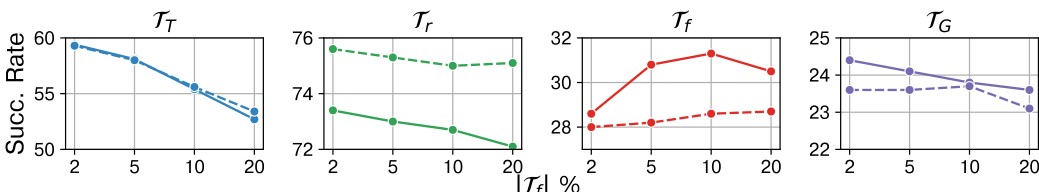

Figure 4: Performance of sequential unlearning on ToolAlpaca.

reinforcing their knowledge. In fact, using the same training data can further strengthen the model's memory on these tools. (c): Preserving General Utility, which improves or maintains the model's general utility through an efficient and effective task arithmetic operation.

## 7 CONCLUSION

We propose Tool Unlearning–a novel machine unlearning task with the goal of unlearning previously learned tools from tool-augmented LLMs. We highlighted the importance of tool unlearning through practical use cases, while also showing why existing unlearning methods are insufficient in this contexts. To systematically address the problem, we develop an effective tool unlearning approach, TOOLDELETE, that enforces three key properties: *tool knowledge removal* for removing any knowledge gained on tools marked for unlearning; *tool knowledge retention* for preserving the knowledge gained on other remaining tools; and *general utility preservation* for maintaining LLM's capability on a range of general tasks like text and code generation. Through extensive experiments conducted on three diverse datasets and with LLMs of two different scales, we demonstrate the effectiveness and efficiency of TOOLDELETE, compared to commonly used general and LLM-specific unlearning approaches. Our results show that TOOLDELETE achieves superior performance by successfully forgetting the tools marked for unlearning.

**Limitations and Future Work**   In this study, we did not conduct experiments using closed-source LLMs or API-based LLMs. Consequently, our findings may not directly extend to such proprietary models, and further research is needed to investigate the applicability of tool unlearning techniques in these contexts. In addition, this work did not investigate the impact of varying model scales due to the limited publicly-available tool-augmented LLMs. Our experiments were conducted on the 7B scale and the scalability of the proposed tool unlearning approach across models of different sizes and scales is an open question for future investigation.

## BROADER IMPACT STATEMENT

Our work investigates the security implications of tool-augmented Large Language Models (LLMs), where we focus on the risks that arise from integrating external tools, and the necessity ability to remove these acquired tools. A key concern is ensuring compliance with privacy regulations, such as the Right to be Forgotten (RTBF), which mandates the removal of specific data upon user request. In the context of tool-augmented LLMs, this necessitates the ability to delete sensitive, regulated, or outdated information related to specific tools. By examining how LLMs interact with and rely on external tools, potential threats to model security can be identified, e.g. unauthorized tool usage, adversarial exploitation, and privacy violations. Our research highlights the critical importance of addressing these challenges.

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

# A  APPENDIX

## A.1  ADDITIONAL RESULTS

We present the results on ToolLLaMA and Gorilla below.

## A.2  IMPLEMENTATION DETAILS

For the checkpoints of tool-augmented LLMs, we used `TangQiaoYu/ToolAlpaca-7B`, `ToolBench/ToolLLaMA-2-7b-v2`, `gorilla-llm/gorilla-openfunctions-v0` that is publically available on Huggingface.

Table 4: Tool unlearning performances when deleting 20% of tools on ToolLLaMA. Evaluation is performed with the specific metric for each tool-augmented LLM on test tools $\mathcal{T}_T$, remaining tools $\mathcal{T}_r$, and unlearned tools $\mathcal{T}_f$, as well as general benchmarks for evaluation LLMs $\mathcal{T}_G$. Best and second best performances are **bold** and underlined respectively.  Original  denotes the tool-augmented LLM prior unlearning and is provided  for reference only .

| Method | $\mathcal{T}_\mathbf{T}(\uparrow)$ | $\mathcal{T}_\mathbf{r}(\uparrow)$ | $\mathcal{T}_\mathbf{f}(\downarrow)$ | General Capability $\mathcal{T}_\mathbf{G}(\uparrow)$ | | | | |
| --- | --- | --- | --- | --- | --- | --- | --- | --- |
| | | | | STEM | Reason | Ins-Follow | Fact | Avg. |
| **Original (Prior Un.)** | 64.0 | 75.6 | 76.0 | 25.3 | 36.8 | 17.3 | 15.0 | 23.6 |
| *General Unlearning Methods* | | | | | | | | |
| RETRAIN | 62.2 | 72.1 | 42.3 | 25.1 | 33.7 | 14.6 | 13.8 | 21.8 |
| GRADASCENT | 42.5 | 56.3 | 51.8 | 14.9 | 26.4 | 11.2 | 8.6 | 15.3 |
| RANDLABEL | 59.3 | 73.5 | 40.7 | 23.4 | 30.6 | 13.3 | 12.7 | 20.0 |
| SALUN | 58.7 | 73.6 | 39.9 | 22.7 | 30.8 | 13.6 | 12.0 | 19.8 |
| *LLM-Specific Unlearning Methods* | | | | | | | | |
| ICUL | 46.2 | 68.2 | 57.2 | 15.1 | 18.8 | 7.1 | 9.4 | 12.6 |
| SGA | 44.7 | 59.6 | 49.4 | 16.3 | 20.4 | 12.8 | 9.7 | 14.8 |
| TAU | 44.5 | 56.3 | 50.2 | 21.6 | 28.0 | 15.3 | 13.5 | 19.6 |
| TOOLDELETE-SFT | 62.8 | 72.8 | 39.5 | 24.6 | 33.4 | 15.8 | 13.7 | **21.9** |
| TOOLDELETE-DPO | **63.2** | **73.6** | **38.7** | 24.3 | 32.9 | 16.0 | 13.8 | 21.8 |

Table 5: Tool unlearning performances when deleting 20% of tools on ToolLLaMA. Evaluation is performed with the specific metric for each tool-augmented LLM on test tools $\mathcal{T}_T$, remaining tools $\mathcal{T}_r$, and unlearned tools $\mathcal{T}_f$, as well as general benchmarks for evaluation LLMs $\mathcal{T}_G$. Best and second best performances are **bold** and underlined respectively. Original denotes the tool-augmented LLM prior unlearning and is provided for reference only .

| Method | $\mathcal{T}_\mathbf{T}(\uparrow)$ | $\mathcal{T}_\mathbf{r}(\uparrow)$ | $\mathcal{T}_\mathbf{f}(\downarrow)$ | General Capability $\mathcal{T}_\mathbf{G}(\uparrow)$ | | | | |
|---|---|---|---|---|---|---|---|---|
| | | | | STEM | Reason | Ins-Follow | Fact | Avg. |
| **Original (Prior Un.)** | 64.0 | 75.6 | 76.0 | 25.3 | 36.8 | 17.3 | 15.0 | 23.6 |
| *General Unlearning Methods* | | | | | | | | |
| RETRAIN | 62.2 | 72.1 | 42.3 | 25.1 | 33.7 | 14.6 | 13.8 | 21.8 |
| GRADASCENT | 42.5 | 56.3 | 51.8 | 14.9 | 26.4 | 11.2 | 8.6 | 15.3 |
| RANDLABEL | 59.3 | 73.5 | 40.7 | 23.4 | 30.6 | 13.3 | 12.7 | 20.0 |
| SALUN | 58.7 | 73.6 | 39.9 | 22.7 | 30.8 | 13.6 | 12.0 | 19.8 |
| *LLM-Specific Unlearning Methods* | | | | | | | | |
| ICUL | 46.2 | 68.2 | 57.2 | 15.1 | 18.8 | 7.1 | 9.4 | 12.6 |
| SGA | 44.7 | 59.6 | 49.4 | 16.3 | 20.4 | 12.8 | 9.7 | 14.8 |
| TAU | 44.5 | 56.3 | 50.2 | 21.6 | 28.0 | 15.3 | 13.5 | 19.6 |
| TOOLDELETE-SFT | 62.8 | 72.8 | 39.5 | 24.6 | 33.4 | 15.8 | 13.7 | **21.9** |
| TOOLDELETE-DPO | **63.2** | **73.6** | **38.7** | 24.3 | 32.9 | 16.0 | 13.8 | 21.8 |

