# OpenReview forum: "Tool Unlearning for Tool Augmented LLMs"
_ICLR.cc/2025/Conference — Submitted to ICLR 2025_

### Official Review · Reviewer_bxXs · 2024-11-02

**Soundness:** 2
**Presentation:** 3
**Contribution:** 2
**Rating:** 5
**Confidence:** 3

**Summary:**

This paper introduces the concept of tool unlearning for tool-augmented large language models (LLMs), addressing the need to remove learned tools due to security, privacy, or obsolescence concerns. Unlike traditional unlearning, tool unlearning focuses on knowledge removal rather than forgetting data samples, involves significant optimization costs, and lacks established evaluation methods. To address these challenges, the authors propose three key properties for effective tool unlearning and present ToolDelete, the first method designed for this purpose. Additionally, they introduce the first membership inference attack (MIA) model to evaluate unlearning effectiveness. Experiments on multiple tool learning datasets demonstrate that ToolDelete can successfully unlearn both randomly chosen tools and specific tool categories without compromising the LLM’s performance on non-deleted tools or general tasks.

**Strengths:**

1.	The problem of tool unlearning is interesting and well-motivated. We definitely need a solution dedicated to achieving tool unlearning for LLM-enabled agents.

2.	The experiment is extensive and comprehensive.

3.	The paper is well-organized and easy for readers to follow.

**Weaknesses:**

1.	I disagree with most differences of tool unlearning compared with sample-level unlearning. According to the entire paper, especially the experiment setups, the tool unlearning also relies on training or tuning with unlearning tool-related data. The model ability regarding certain tasks or tools is reflected in the data samples, in this sense, sample-level unlearning and tool unlearning are the same. Except for the difference regarding Evaluation, I don’t think there is any uniqueness in tool unlearning.

2.	Two terms of Eq. (5) require the base model f_0 without learning any tools. However, in practical scenarios, only the model f that has learned many tools is available to unlearn certain tools. How does ToolDelete work in the case when f_0 is unavailable?

3.	In terms of LiRA-Tool, how you generate ‘shadow samples’ via prompting GPT-4, detailed instructions or prompts should be provided.

4.	A comparison between traditional sample-level LiRA and LiRA-Tool is needed, which can reflect the issues of sample-level LiRA mentioned in Lines 296-298.

5.	The training time comparison to retraining from scratch has no significance. The comparison between ToolDelete and other baseline unlearning approaches is more needed.

**Questions:**

See the weaknesses above

---

> ### Author Response · Authors · 2024-11-26
>
> **Re W1: I disagree with most differences of tool unlearning compared with sample-level unlearning. According to the entire paper, especially the experiment setups, the tool unlearning also relies on training or tuning with unlearning tool-related data. The model ability regarding certain tasks or tools is reflected in the data samples, in this sense, sample-level unlearning and tool unlearning are the same. Except for the difference regarding Evaluation, I don’t think there is any uniqueness in tool unlearning.**
>
> As we explained in Line 135-153, tool unlearning differs from traditional unlearning in several aspects, specifically objective, evaluation, and data.
>
> Unlike traditional unlearning, which focuses on individual data samples, tool unlearning targets the removal of parametric knowledge associated with an entire tool’s functionality. This requires more than isolated data deletion—it must balance forgetting targeted tools without impacting remaining tools.
>
> Evaluation also differs, as tool unlearning assesses whether the model can still utilize specific tools rather than remembering specific examples or relying on metrics like extraction probability.
>
> Furthermore, traditional methods often need access to exact training data, which may not be available for tool unlearning, in particular, in closed-source settings.
>
> Finally, our results in Table 1 clearly demonstrate that traditional unlearning methods fall short in tool unlearning.
>
> These differences make tool unlearning a distinct and more complex task.
>
>
> **Re W2: Two terms of Eq. (5) require the base model f_0 without learning any tools. However, in practical scenarios, only the model f that has learned many tools is available to unlearn certain tools. How does ToolDelete work in the case when f_0 is unavailable?**
>
> The goal of $f_0$ is to provide the response of an LLM without learning $T_f$. In the case that $f_0$ is missing, we can use a randomly initialized LLM in place of $f_0$, because both $f_R$ and $f_0$ have not been trained on $T_f$.
>
>
> **Re W3: In terms of LiRA-Tool, how you generate ‘shadow samples’ via prompting GPT-4, detailed instructions or prompts should be provided.**
>
> Below is an example of prompting GPT-4 to generate shadow samples.
>
> >You are now a synthetic data generator. Generate query-response pairs to evaluate an LLM's ability of using an API.
>
> >How to generate "query": Based on the API and documentation shown below, think of a user query that needs to be answered by calling the API.
>
> >How to generate "response": Write down the correct API call with correct arguments.
>
> >The in-context examples below demonstrate what you need to generate. Please be as diverse and creative as possible in phrasing and style. But do not hallucinate.
>
> >## In-context Examples
> >#### Tool and Documentation
> >Name: StableDiffusionPipeline.from_ pretrained ()
>
> >#### Query
> >I want to see some cats dancing in celebration!
>
> >#### Response
> >API call: StableDiffusionPipelin e.from_pretrained(stabilityai/stable-diffusion-2-1)
>
>
> >Now, for the following API, generate a query-response pair.
>
> >#### Tool and Documentation
> >api_name()
>
> >#### Query
>
> >#### Response
>
>
> We will release the prompt and the samples, as well as our code to generate that upon acceptance.
>
>
> **Re W4: A comparison between traditional sample-level LiRA and LiRA-Tool is needed, which can reflect the issues of sample-level LiRA mentioned in Lines 296-298.**
>
> The original LiRA evaluates whether a specific sample is part of the training data by approximating the in-training and out-of-training distributions using shadow models. This approach is insufficient (and potentially inaccurate and misleading) for tool unlearning, where the focus is on removing knowledge related to an entire tool rather than individual data points.
>
>
>
> **Re W5: The training time comparison to retraining from scratch has no significance. The comparison between ToolDelete and other baseline unlearning approaches is more needed.**
>
> We present the time when deleting 2% of tools comparison to baselines below:
> |            | Time (hrs) |
> |------------|------------|
> | Retrain    | 6.7        |
> | GradAscent | 0.3        |
> | RandLabel  | 2.3        |
> | SalUn      | 2.4        |
> | ICLU       | NA         |
> | SGA        | 3.7        |
> | TAU        | 3.7        |
> | ToolDelete | 0.4 |
>
> ICLU uses prompt-based methods, which imposes offline cost. ToolDelete is more efficient than RandLabel and SalUn. ToolDelete consumes more time than GradAscent, SGA and TAU. However, the efficiency of these three models is at the cost of considerably poor unlearning performances, see Table 1.

---

> > ### Comment · Reviewer_bxXs · 2024-11-29
> >
> > Thanks for the response. After careful reading, I still have the following concerns:
> >
> > 1. Regarding the difference between Tool unlearning and regular sample unlearning, your response reframes what you wrote in the paper in my view. I would like to see deeper explanations or examples. Moreover, suppose you claim that there is distinctness in terms of data. In that case, you should mention what modification, design, and preparation you made on the data when you want to unlearn certain knowledge of a certain tool rather than directly using a well-built dataset (ToolAlpaca) to implement such unlearning. As ToolAlpaca is collected with the assumption of user historical usage, if you directly conduct unlearning on these samples, your unlearning is sample-level. As you mentioned, traditional methods need access to exact training data, your method does not, but you should describe what data you need and how to obtain the data (this part also needs to be reflected in the experiments).
> >
> > 2. If $f_0$ can be a random initialized LLM, you should present the experiment results.
> >
> > 3. You mentioned that sample-level LiRA may be potentially insufficient, inaccurate, and misleading in Tool unlearning cases, you should conduct experiments to validate this, which can also demonstrate your novelty of modifying LiRA into LiRA-tool.
> >
> > 4. Your statement 'ToolDelete consumes more time than GradAscent, SGA, and TAU' seems conflict to with the table you present.

---

> > > ### Author Response · Authors · 2024-12-01
> > > **Response - Part 1**
> > >
> > > Thank you for your feedback and engagement in discussion.
> > >
> > > **Re W1: Difference between Tool unlearning and regular sample unlearning:**
> > >
> > >
> > > **Example and deeper explanation on the difference between tool unlearning and sample-level unlearning in objective and evaluation**
> > >
> > > In Figure 1, we provided an example to illustrate the key distinctions between tool unlearning and sample-level unlearning. Tool unlearning fundamentally targets the removal of parametric knowledge related to an entire tool, rather than forgetting specific training samples (e.g. decreasing lexical memorization of query-response pairs of a tool). For instance, unlearning an API tool requires removing the capability to solve tasks using the API, regardless of the presence or absence of specific training samples. Therefore, the model must be able to unlearn even examples that were not explicitly part of the training data. Please see further analysis below.
> > >
> > >
> > > **Difference in data**
> > >
> > > To further address the reviewer’s comment, we conducted additional experiments, where we assumed having **no** access to the **exact** training samples. In this case, we can generate pseudo samples, i.e. query-response pairs, for a given tool, using a similar method as generating shadow samples in LiRA-Tool (it is important to note that these are newly generated samples, not the same as the shadow samples used in evaluation).
> > >
> > > The results indicate minimal performance degradation when exact training samples are unavailable. This indicates the robustness of ToolDelete and highlights a key difference between tool unlearning and sample-level unlearning in terms of access to the training data. ToolDelete can use any appropriate data to unlearn the skill of using tools.
> > >
> > > | Method           | $T_T (\uparrow)$ | $T_r (\uparrow)$ | $T_f (\downarrow)$ | $T_G (\uparrow)$ |
> > > |------------------|------------------|------------------|--------------------|------------------|
> > > |W/o access to training samples|
> > > | GradAscent       | 35.6             | 48.3             | 38.2               | 13.5             |
> > > | RandLabel        | 48.4             | 71.5             | 39.4               | 19.8             |
> > > | SalUn            | 48.5             | 70.8             | 33.7               | 21.3             |
> > > | ICUL             | 48.1             | 70.3             | 68.5               | 10.8             |
> > > | SGA              | 43.2             | 61.5             | 39.7               | 17.3             |
> > > | TAU              | 44.7             | 61.6             | 38.8               | 21.1             |
> > > | ToolDelete - SFT | _52.0_           | _72.5_           | _30.1_             | **22.8**         |
> > > | ToolDelete - DPO | **52.9**         | **76.0**         | **28.0**           | _22.5_           |
> > > |------------------|------------------|------------------|--------------------|------------------|
> > > |W/ access to training samples|
> > > |------------------|------------------|------------------|--------------------|------------------|
> > > | ToolDelete - SFT | _52.7_           | _72.1_           | _30.5_             | **23.6**         |
> > > | ToolDelete - DPO | **53.4**         | **75.1**         | **28.7**           | _23.1_           |
> > >
> > >
> > >
> > > **Re W2: If $f_0$ can be a random initialized LLM, you should present the experiment results.**
> > >
> > > The following Table shows the results when using a randomly initialized LLM $f_R$ in place of $f_0$. Results below show that both $f_R$ and $f_0$ can achieve effective tool unlearning and outperforms baselines; $f_R$ achieves better performance in forgetting tool knowledge, while $f_0$ can better preserve tool knowledge.
> > >
> > > | Method           | $T_T (\uparrow)$ | $T_r (\uparrow)$ | $T_f (\downarrow)$ | $T_G (\uparrow)$ |
> > > |------------------|------------------|------------------|--------------------|------------------|
> > > | ToolDelete - SFT, using $f_0$ | _52.7_           | _72.1_           | _30.5_             | **23.6**         |
> > > | ToolDelete - DPO, using $f_0$  | **53.4**         | **75.1**         | **28.7**           | _23.1_           |
> > > |------------------|------------------|------------------|--------------------|------------------|
> > > | ToolDelete - SFT, using $f_R$ | _50.9_           | _71.3_           | _29.8_             | **22.7**         |
> > > | ToolDelete - DPO, using $f_R$  | **52.6**         | **73.4**         | **27.5**           | _22.4_           |
> > >
> > > - On $T_T$ (performance on test tool) and $T_r$ (performance on remaining tool): $f_0$ achieves higher scores, which indicates its advantage in generalization.
> > > - On $T_f$ (performance on tools marked for unlearning): $f_R$ performs better and effectively forgets tool-specific knowledge.
> > > - On $T_G$ (performance on general benchmarks for LLM evaluation) $f_0$ consistently achieves higher scores for overall generalization.
> > >
> > > These results highlight the flexibility of ToolDelete, whether leveraging pretrained or randomly initialized models.

---

> > > > ### Author Response · Authors · 2024-12-01
> > > > **Response - Part 2**
> > > >
> > > > **Re W3: You mentioned that sample-level LiRA may be potentially insufficient, inaccurate, and misleading in Tool unlearning cases, you should conduct experiments to validate this, which can also demonstrate your novelty of modifying LiRA into LiRA-tool.**
> > > >
> > > > The fundamental reason of preferring LiRA-Tool (diverse prompts) over sample-level LiRA (limited prompts) is because of the experiments we showed above: **unlearning samples in the training set does not necessarily mean successful unlearning of tool knowledge.**
> > > >
> > > > In fact, LiRA only measures losses on the training samples, which may not be comprehensive or diverse enough. Merely relying on these samples is insufficient to evaluate if tool knowledge has been truly removed from the parametric knowledge of an LLM.
> > > >
> > > > In contrast, LiRA-Tool addresses these limitations by stress-testing the LLM with diverse prompts, without “overfitting” to the limited training samples. Therefore, we can be more confident about whether the knowledge removal is successful or not.
> > > >
> > > > Direct comparison between LiRA and LiRA-Tool is not possible because they evaluate models on different sets of samples. However, a proxy approach is to use the success rate on $T_f$, which directly reflects the tool knowledge of a model as a proxy for comparison, i.e. we expect a good metric to have higher correlation with $T_f$. The results are shown in the following Table; the Pearson correlation between $T_f$ and LiRA-Tool is 0.44, while the corresponding value for LiRA is -0.04, which indicates LiRA-Tool better correlates with performance on $T_f$.
> > > >
> > > > In addition, in case of Retrain specifically (which is expected to have minimal knowledge on tools marked for unlearning), we see that LiRA-Tool shows better knowledge removal (lower value, 0.16), while the original LiRA shows worse performance (higher value, 0.20), indicating LiRA-Tool better score Retrain (as the oracle).
> > > >
> > > > | Method           | $T_f (\downarrow)$ | LiRA-Tool $(\downarrow)$ | Original LiRA $(\downarrow)$ |
> > > > |------------------|--------------------|--------------------------|------------------------------|
> > > > | Retrain          | 38.5               | 0.16                     | 0.20                         |
> > > > | GradAscent       | 34.6               | 0.22                     | 0.11                         |
> > > > | RandLabel        | 37.5               | 0.17                     | 0.14                         |
> > > > | SalUn            | 38.2               | 0.17                     | 0.14                         |
> > > > | ICUL             | 58.3               | 0.19                     | 0.12                         |
> > > > | SGA              | 42.1               | 0.20                     | 0.13                         |
> > > > | TAU              | 42.5               | 0.20                     | 0.13                         |
> > > > | ToolDelete - SFT | _30.5_             | _0.16_                   | 0.12                         |
> > > > | ToolDelete - DPO | **28.7**           | **0.14**                 | 0.13                         |
> > > >
> > > >
> > > >
> > > >
> > > > **Re W4: time efficiency**
> > > >
> > > > We thank the reviewer for pointing out the typo.
> > > >
> > > > To clarify, ToolDelete needs more training time (0.1 hrs) than GradAscent, but GradAscent’s low training cost is at the cost of significant performance degradation, making it not practical.
> > > >
> > > > ToolDelete is more time-efficient than SGA and TAU (saves 3.3 hrs). This is because these two methods need to measure the memorization of samples periodically, which adds considerable overhead.

---

> > > > > ### Comment · Reviewer_bxXs · 2024-12-02
> > > > >
> > > > > Thanks for the further reply. I would like to increase my rating to 5.

---

> > > > > > ### Author Response · Authors · 2024-12-02
> > > > > >
> > > > > > Thank you for your reply and increasing the score to 5.
> > > > > >
> > > > > > Is there any concern that is still unresolved? We would be more than happy to provide further clarifications and details, or conduct additional experiments (if time permits).
> > > > > >
> > > > > > Thank you.

---

### Official Review · Reviewer_hnq1 · 2024-11-03

**Soundness:** 3
**Presentation:** 3
**Contribution:** 3
**Rating:** 5
**Confidence:** 3

**Summary:**

The authors introduce a new kind of task, "tool forgetting". Compared with traditional forgetting, it needs to deal with different challenges: it is knowledge to be removed rather than individual samples to be forgotten, and the cost of optimizing LLMs is high. To address these issues, the authors propose TOOLDELETE, and a member inference attack model LiRA-Tool, .

**Strengths:**

(1)The tool forgetting task is innovatively proposed, opening up a new direction for related research.

(2) The TOOLDELETE method has a clear design and implementation, and the experimental results show that it is effective in tool forgetting.

(3) A LiRA-Tool model is developed to evaluate tool forgetting, which is novel.

**Weaknesses:**

(1) Does the author's approach apply to closed-source or API-only LLMs? If so, the author can perform some extended experiments to better illustrate the effect of TOOLDELETE.

(2) There is a lack of sufficient research on the impact of different model sizes on tool forgetting.

(3) The shadow samples in LiRA-Tool may not fully represent the complexity of the original tool learning data, which affects the accuracy of the evaluation.

**Questions:**

The author needs to explain in detail the questions I mentioned above, and I will determine the final score based on your answers.

---

> ### Author Response · Authors · 2024-11-26
>
> **Re W1: Does the author's approach apply to closed-source or API-only LLMs? If so, the author can perform some extended experiments to better illustrate the effect of TOOLDELETE.**
>
> Similar to most existing unlearning approaches, our method is currently applicable to only open-source LLMs, which comprise the majority of tool-augmented LLMs. Unlearning from close-source models is indeed an important area, which is the subject of our future work.
>
>
> **Re W2: There is a lack of sufficient research on the impact of different model sizes on tool forgetting.**
>
> Due to the limited available tool-augmented LLMs at different scales, we were only able to conduct experiments on LLMs of a maximum of 7B parameters. Conducting additional experiments can be easily achieved, if there exists tool-augmented LLMs of different scales. We would be eager to include such analysis in the final version of this work.
>
>
> **Re W3: The shadow samples in LiRA-Tool may not fully represent the complexity of the original tool learning data, which affects the accuracy of the evaluation.**
>
> Yes, we acknowledge that the shadow samples may not fully represent the complexity of the original tool learning data. However, the idea is to not overfit to the training prompt, and evaluate the tool-using ability with diverse prompts. Given the diversity of prompts, we aim to achieve a more robust evaluation that provides higher confidence than simply using the training samples.

---

> > ### Comment · Reviewer_hnq1 · 2024-12-03
> >
> > Thank you to the author for your response. However, I don't think the author responded well to my second question, so I decided to lower my score to 5.

---

> > > ### Author Response · Authors · 2024-12-03
> > >
> > > We thank the reviewer for their feedback.
> > >
> > > Could you please be specific about what concern is unresolved? We are more than happy to provide more explanations.
> > >
> > > Thank you.

---

### Official Review · Reviewer_EqGf · 2024-11-03

**Soundness:** 3
**Presentation:** 4
**Contribution:** 2
**Rating:** 5
**Confidence:** 4

**Summary:**

The paper propose a novel threat model that looks at tool use knowledge as source of privacy and security issues and therefore proposes methods to unlearn them. One of the more convincing examples is that a tool that knows how to operate a private tool might reveal to adversary some information.

The paper goes with proposing a ToolDelete method that using three separate objectives: removal, retention and utility retention finetunes/DPO the model to achieve these objectives. The paper further states challenges with evaluation and adapts LiRA approach to measure method efficiency.

**Strengths:**

- Overall the paper is well written with a standard and comprehensive evaluation and methods.
- The paper presents a novel threat model
- Evaluation focuses on three popular benchmarks for tool use
- Results show a strong advantage of the proposed ToolDelete method over standard unlearning tools

**Weaknesses:**

- It is not clear if the threat model is valid -- the right to be forgotten claims that appear in intro and broader impacts have no relation to the tool use and therefore invalid. A credible attack that shows privacy or security issues is required to justify the proposed problem. While the problem itself is novel I struggle to see the value in addressing this problem, i.e. why would anybody want to unlearn tools for privacy/security reasons.

- It is mentioned that full re-training is costly and pointless, however while this argument is relevant  for unlearning literature in general it does not look to me relevant for this particular problem. As the pipeline commonly shows tool use training comes after the most costly part of model pre-training and therefore is not that challenging to do. Furthermore, it is not clear that models should be trained to support all tool use in practice -- it could be that for particular use cases the company would only train for a particular sub-set of tools and therefore retraining the model is not a challenge.

-- novelty: while the threat model is novel it is not clear, how novel is the contribution as the proposed method simply balances three losses that seems like a rather straightforward way to build the tooldelete method -- explaining the challenges would be interesting. However, I have to note LiRA-Tool adaptation is nice trick (nevertheless the paper does not focus on this part as much).

-- Investigating imbalance in tool unlearning seems interesting but does not exist in the paper. The authors only measure percentage of unlearned tools, but I would rather see how different tools impact unlearning metrics --> where does the drop in Table 1 come from?

**Questions:**

- explain the threat model
- differentiate more from naive solutions
- provide more justification on novelty of the approach
- investigate how different tools are unlearned

---

> ### Author Response · Authors · 2024-11-26
> **Response - Part 1**
>
> **Re W1: It is not clear if the threat model is valid -- the right to be forgotten claims that appear in intro and broader impacts have no relation to the tool use and therefore invalid. A credible attack that shows privacy or security issues is required to justify the proposed problem. While the problem itself is novel I struggle to see the value in addressing this problem, i.e. why would anybody want to unlearn tools for privacy/security reasons.**
>
> As we described in Line 127-133, real-world application of tool unlearning include addressing the insecure tools from untrustworthy developers that could be exploited by adversarial attackers, removing tools restricted by their providers due to copyright or privacy concerns, such as APIs that start allowing unauthorized downloads of book chapters or releasing publications that users did not author, and may others described in the paper. These real world cases are covered in API Bench, ToolBench, and ToolAlpaca datasets used in our experiments.
>
> In addition, the principles of "Right to Be Forgotten" (RTBF) “involves revoking public access to information that was known publicly at a certain time,” according to wikipedia. Developers of tools or users who contribute their data for tool creation can request their data to be deleted.
>
>
> **Re W2: It is mentioned that full re-training is costly and pointless, however while this argument is relevant for unlearning literature in general it does not look to me relevant for this particular problem. As the pipeline commonly shows tool use training comes after the most costly part of model pre-training and therefore is not that challenging to do. Furthermore, it is not clear that models should be trained to support all tool use in practice -- it could be that for particular use cases the company would only train for a particular sub-set of tools and therefore retraining the model is not a challenge.**
>
> Tool augmentation is indeed less expensive than pretraining. However, it can still be an expensive and resource-intensive process. As shown in Figure 3, ToolDelete saves an average of 74.8% in training time (reducing turnaround time from 6.7 hrs to 0.42 hrs when unlearning 2% tools, i.e. an efficiency gain of 6.3 hours) compared to retraining from scratch, based on experiments conducted on 4 A100 GPUs. Therefore, ToolDelete is a more practical and cost-effective approach. This is especially important when tools need to be unlearned dynamically or when computational resources or time are limited.
>
>
> **Re W3: Novelty: while the threat model is novel it is not clear, how novel is the contribution as the proposed method simply balances three losses that seems like a rather straightforward way to build the tooldelete method -- explaining the challenges would be interesting. However, I have to note LiRA-Tool adaptation is nice trick (nevertheless the paper does not focus on this part as much).**
>
> To the best of our knowledge, we are the first to conceptualize and study the question of tool unlearning from tool-augmented LLMs. The novelty of ToolDelete is in addressing the unique challenges of tool unlearning, which go beyond straightforward optimization: targeted removal without over-forgetting, scalability and efficiency, and dynamic and sequential unlearning. In addition, we proposed and analyzed LiRA-Tool as an effective membership inference attack to assess tool unlearning.
>
>
> **Re W4: Investigating imbalance in tool unlearning seems interesting but does not exist in the paper. The authors only measure percentage of unlearned tools, but I would rather see how different tools impact unlearning metrics --> where does the drop in Table 1 come from?**
>
> Since this is the first investigation on tool unlearning, we did not investigate the finer impacts of different types of tools in unlearning, and defer it to future work. We acknowledge that investigating the imbalance in tool unlearning and how different tools affect metrics is an important direction for deeper analysis. In Table 1, the performance drop can stem from several factors related to tool-specific characteristics, e.g. tool complexity, frequency in training, potential overlapping representations between tools.

---

> > ### Author Response · Authors · 2024-11-27
> > **Response - Part 2**
> >
> > **Re Q1: explain the threat model**
> >
> > The threat model is a tool augmented LLM $f$ that has been tuned on tool-augmentation dataset $D = \{\mathcal{T}, \mathcal{Q}, \mathcal{Y} \}$. Part of the tools $\mathcal{T}_f \in \mathcal{T}$ may be requested to unlearn from $f$. The goal of tool unlearning is to obtain an unlearned model $f’$ such that $f’$ does not possess knowledge on $\mathcal{T}_f$.
> >
> >
> > **Re Q2: differentiate more from naive solutions**
> >
> > The naive solutions such as retraining or using traditional unlearning approaches may seem straightforward, but they fall short in several key aspects compared to our method. This is confirmed by our main results in Table 1. The main reason is that the naive solutions are designed for sample-level unlearning, while ToolDelete is designed for knowledge-level tool unlearning, discussed in Section 3 Lines 135-153.
> >
> >
> > **Re Q3: provide more justification on novelty of the approach**
> >
> > The novelty of ToolDelete is in addressing the unique challenges of tool unlearning: targeted removal without over-forgetting, scalability and efficiency, and dynamic and sequential tool unlearning. In addition, we proposed and analyzed LiRA-Tool as an effective membership inference attack to assess tool unlearning.
> >
> > **Re Q4: investigate how different tools are unlearned and their effect**
> >
> > We conduct additional experiments on class-wise unlearning, where tools to unlearn $T_f$ belong to the same category. These results can compare how different tools affect unlearning performances and uncover how different tools are unlearned.
> >
> > When unlearning tools tagged as “Development” tools:
> >
> > | Method           | $T_T (\uparrow)$ | $T_r (\uparrow)$ | $T_f (\downarrow)$ | $T_G (\uparrow)$ |
> > | NPO              | 46.1             | 70.4             | _29.8_             | 18.4             |
> > | CUT              | 42.4             | 67.8             | 32.8               | 18.0             |
> > | ToolDelete - SFT | _48.3_           | _71.4_           | 30.4               | _20.8_           |
> > | ToolDelete - DPO | **50.5**         | **72.6**         | **29.5**           | **22.3**         |
> >
> > When unlearning tools tagged as “Transportation” tools:
> >
> > | Method           | $T_T (\uparrow)$ | $T_r (\uparrow)$ | $T_f (\downarrow)$ | $T_G (\uparrow)$ |
> > |------------------|------------------|------------------|--------------------|------------------|
> > | NPO              | 44.7             | 70.3             | _28.1_             | 18.1             |
> > | CUT              | 41.6             | 69.5             | 33.6               | 17.7             |
> > | ToolDelete - SFT | _48.8_           | _71.0_           | 29.8               | **_21.4_**       |
> > | ToolDelete - DPO | **49.2**         | **73.3**         | **27.4**           | **_20.7_**       |
> >
> > We notice that unlearning “Development” tools is more challenging than unlearning. The reason is that if a tool-augmented LLM unlearns Development tools, the LLM may be required to forget more knowledge on using tools, which influences the knowledge on other tools.
> >
> > We will include more class-wise unlearning in the camera-ready version.

---

### Official Review · Reviewer_Nztc · 2024-11-03

**Soundness:** 3
**Presentation:** 3
**Contribution:** 3
**Rating:** 6
**Confidence:** 3

**Summary:**

This paper introduces a "tool unlearning" method for tool-augmented large language models (LLMs), which effectively removes the capability to use specific tools without compromising the model's overall abilities. The study presents the TOOLDELETE algorithm, which successfully erases targeted tool knowledge while retaining the model’s performance in using other tools and general tasks. The LiRA-Tool evaluation method confirms the effectiveness of this approach. This technique is significant for meeting data privacy and compliance requirements.

**Strengths:**

* Propose the novel task tool unlearning and describe the difference between this and traditional sample-level unlearning.

* Make concerns about unlearning knowledge of targeted tools, retaining knowledge of other tools and general ability of LLMs.

* Main experiments show good performance in unlearning and retaining knowledge of the proposed method.

**Weaknesses:**

* Clear real-world applications cannot be found. This paper provides a real-world harmful case: unlearning tool-augmented LLM with knowledge of insecure HTTP requests. However, it is hard to find the applications that need the proposed tool unlearning. The authors should clearly demonstrate the practice of the proposed approach in real-world applications. Firstly, this case does not match the authors' claim of "preserving its ability to use others tools and perform general tasks". Secondly, as described in the paper, LLMs retain "harmful knowledge" from tools, and this knowledge may be unlearned through a traditional unlearning process(constructing a dataset of question-answer pairs). The author must prove why we need tool unlearning rather than traditional unlearning in open-world cases.

* Lack of description of loss function design and mistakes found in section 4.4. As shown in the authors' objective function, the design of the function '$g(f, t)$' is of great importance and directly decides whether the tool unlearning can succeed or not. However, the authors do not report details of the function $g$. This makes it hard to evaluate the effectiveness of the method. Thus, in Eq. (5), $[g(f_0, t_i) − g(f′, t_i)]$ and $[g(f, t_m) − g(f′, t_m)]$, as far as I'm concerned, lead to a certain score, while $\alpha(\theta_0 − \theta_R)$ leads to a vector. These cannot be added together. The authors should elaborate more on each term of the loss function.

* The author claims that their method can keep the knowledge from other tools after the tool unlearning process. However, they do not conduct experiments on this. I would like the authors to conduct more experiments on an LLM with two or more tools' augmentation to prove the claimed benefits.

* Further suggestions: more mainstream datasets and benchmarks can be utilized in the experiment, e.g., [1], [2].

* Typos: "preservemode's" in line 215 on page 5.

[1] Li, Minghao, et al. "Api-bank: A comprehensive benchmark for tool-augmented LLMs." arXiv preprint arXiv:2304.08244 (2023).
[2] Zhuang, Yuchen, et al. "Toolqa: A dataset for LLM question answering with external tools." Advances in Neural Information Processing Systems 36 (2023): 50117-50143.

**Questions:**

* Why do we really need tool unlearning rather than traditional unlearning in open-world cases?

The author have made good efforts in solving my problems. Thus, I will raise my score to 6.

---

> ### Author Response · Authors · 2024-11-26
> **Response - Part 1**
>
> **Re W1: Clear real-world applications cannot be found. This paper provides a real-world harmful case: unlearning tool-augmented LLM with knowledge of insecure HTTP requests. However, it is hard to find the applications that need the proposed tool unlearning. The authors should clearly demonstrate the practice of the proposed approach in real-world applications. Firstly, this case does not match the authors' claim of "preserving its ability to use others tools and perform general tasks". Secondly, as described in the paper, LLMs retain "harmful knowledge" from tools, and this knowledge may be unlearned through a traditional unlearning process(constructing a dataset of question-answer pairs). The author must prove why we need tool unlearning rather than traditional unlearning in open-world cases.**
>
> (a) Real-world applications
>
> As we show in Section 3 Line 127-133, real-world application of tool unlearning include:
>
> - addressing the insecure tools from untrustworthy developers that could be exploited by adversarial attackers
> - removing tools restricted by their providers due to copyright or privacy concerns, such as APIs that start allowing unauthorized downloads of book chapters or releasing publications that users did not author
> - unlearning broken or deprecated tool that lead to failed operations or corrupted outputs
> - unlearning tools that may no longer be needed
> - managing limited model capacity, where new tools and evolving needs necessitate replacing outdated tools.
>
> These real world cases are covered in API Bench, ToolBench, and ToolAlpaca datasets used in our experiments.
>
> (b) “preserving its ability to use other tools and perform general tasks"
>
> This statement reflects a core requirement in tool unlearning—the unlearning process should target the specific tools marked for unlearning without negatively affecting LLM’s ability to use the other remaining tools. This principle applies in general, regardless of the downstream applications. In the HTTP example, the goal is to unlearn insecure API usage, while retaining LLM's ability to work with secure APIs and general-purpose capabilities like text generation or instruction-following.
>
> (c) Why are traditional unlearning approaches not applicable to tool unlearning?
>
> As we explained in Line 135-153, tool unlearning differs from traditional unlearning in several aspects, specifically objective, evaluation, and data. Unlike traditional unlearning, which focuses on individual data samples, tool unlearning targets the removal of parametric knowledge associated with an entire tool’s functionality. This requires more than isolated data deletion—it must balance forgetting targeted tools without impacting remaining tools. Evaluation also differs, as tool unlearning assesses whether the model can still utilize specific tools rather than remembering specific examples or relying on metrics like extraction probability. Furthermore, traditional methods often need access to exact training data, which may not be available for tool unlearning, in particular, in closed-source settings. Finally, our results in Table 1 clearly demonstrate that traditional unlearning methods fall short in tool unlearning. These differences make tool unlearning a distinct and more complex task.
>
> **Re W2: Lack of description of loss function design and mistakes found in section 4.4. As shown in the authors' objective function, the design of the function '$g(f, t)$' is of great importance and directly decides whether the tool unlearning can succeed or not. However, the authors do not report details of the function. This makes it hard to evaluate the effectiveness of the method. Thus, in Eq. (5), $[g(f_0, t_i) - g(f’, t_i)]$ and $[g(f_0, t_m) - g(f’, t_m)]$, as far as I'm concerned, lead to a certain score, while $\alpha(\theta_0 - \theta_R)$ leads to a vector. These cannot be added together. The authors should elaborate more on each term of the loss function.**
>
> The idea of $g(f, t)$ is to measure the knowledge of a model on specific tools, which, in our case, is the task (language modeling) loss.
>
> Equation 5 consists of two parts.
>
> 1. $\arg\min_{\theta_{f'}} E\{[g(f_0, t_i) - g(f’, t_i)] + [g(f_0, t_m) - g(f’, t_m)]\}$   The first part is an optimization problem, where we search for the model parameters that minimize the loss $[g(f_0, t_i) - g(f’, t_i)]$ and $[g(f_0, t_m) - g(f’, t_m)]$. The output is the model parameters ($\theta_{f’}$), not a score.
> 2. The second part $\alpha(\theta_0 - \theta_R)$ is a vector of the same size, which is applied to the solution of the optimization above.
>
> Therefore, combining two will lead to the final model parameters.

---

> > ### Comment · Reviewer_Nztc · 2024-11-27
> > **Reply to authors - Part 1**
> >
> > Thank you all for your response!
> >
> > For W1, (a) (b), actually I want a real-world scenario that exactly shows the necessarity of your method. Could you offer one?
> >
> > In W1, (c), I notice that you mention "traditional methods often need access to exact training data, which may not be available for tool unlearning, in particular, in closed-source settings". Can closed-source models be unlearned? For open-source LLMs, why can't we use traditional methods? I doubt this.
> >
> > For W2, I want to see what function g exactly is. And that will help my judgement a lot.

---

> > > ### Author Response · Authors · 2024-11-27
> > > **Response to Reviewer's Reply**
> > >
> > > We thank the reviewer for their fast response and engagement in discussion.
> > >
> > > **Re W1 (a)(b): real-world applications of ToolDelete**
> > >
> > > Before answering the reviewer’s question, we would like to describe how tool-augmented LLMs are developed and trained. Tool-augmented LLMs are fine-tuned from LLMs using tool-using datasets, which consists of instruction-response pairs. Training embeds the ability to use tools / APIs into the LLM’s "parametric knowledge". We acknowledge that there are tool LLMs that have retrieval systems or are augmented from frozen LLMs. But in our experiments, we focus on tool LLMs that are explicitly tuned to use tools.
> > >
> > > Given the above context, we provide several examples in which tool unlearning is essential:
> > >
> > > 1. Imagine a tool-augmented LLM that is deployed in a healthcare system and trained to use APIs for handling and processing patient data, such as accessing medical records or generating anonymized reports. Suppose one of the APIs that was initially compliant is later flagged as insecure due to a vulnerability that could expose patient data. This violates regulations like HIPAA or GDPR. In this case, ToolDelete is essential as it can update the tool-augmented LLM's parameters to unlearn how to invoke the insecure API. This removes any capability embedded in the LLM's parametric knowledge and prevents adversarial or accidental usage of the vulnerable API.
> > >
> > > 2. Consider a tool-augmented LLM that can use a Safe-For-Work diffusion model as a tool to generate images based on user instructions. If the user prompts can fool the model to generate Not Safe For Work, harmful, or biased images, this tool should be unlearned from the LLM. Note that even if we augment the LLM with a new and safe version of the diffusion model without unlearning the previous version, the LLM would still be able to call the previous version, which can lead to generating Not Safe For Work, harmful, or biased images. Therefore, we should explicitly erase the ability of using the previous version of the diffusion model from the LLM.
> > >
> > > 3. Tool unlearning is also essential when a tool has a major update, where the function names and input parameters have changed, e.g. the major update of the Python transformers package from v2 to v4. Without unlearning v2, the tool-augmented LLM may generate erroneous code and bring difficulty for debugging, since many functions have been renamed and removed. Therefore, as the underlying tools evolve, the tool-augmented LLM should be updated through unlearning of the previous versions and augmenting the new ones.
> > >
> > >
> > > **Re W1 (c): traditional methods**
> > >
> > > We highlight that a major difference between traditional unlearning and tool unlearning is in the need to access exact training data.
> > >
> > > For open-source LLMs, we can use traditional methods. However,  they focus on unlearning individual data samples, as opposed to tool unlearning, which targets the removal of parametric knowledge associated with an entire tool’s functionality. Our results in Table 1 also confirm that the traditional models underperform in case of tool unlearning.
> > >
> > > For closed-source LLMs, tools can be unlearned by model owners who have access to the model’s parameters and training environment.
> > >
> > > **Re W2: exact for of $g$**
> > > $g(f, t)$ computes the average language modeling loss over all query-response pairs for tool $t$ with LLM $f$.  The formula is
> > >
> > > $g(f, t) = \frac{1}{|t|}\sum_{y \in t}L_{LM}(f, y) = \frac{1}{|t|}\sum_{i}^N log P(y_i | y_{<i}; f)$,
> > >
> > > where $|t|$ is the number of demonstrations of tool $t$, $N$ denotes number of tokens in sequence $y$, $y_{<i}$ denotes tokens prior to token $y_i$.

---

> > > > ### Comment · Reviewer_Nztc · 2024-11-28
> > > > **Response to authors**
> > > >
> > > > Thank you all for your efforts!
> > > > Your explanation of real-world application solved my problem.
> > > >
> > > > For W1 (c), actually, for those who have access to the model’s parameters and training environment,  the model is open-source. That is, in my opinion, your method can be applicated in a white-box setting. If you agree with me, you can mention in your paper to make readers understand better.
> > > >
> > > > For W2, seeing your function g, I can guess that you measure the knowledge of LLMs with tools through a list of question-answer pairs. I want to know what categories of pairs can be found in this list. And, in supervised finetuning or DPO, as far as I'm concerned, the loss is an exact number, for each question & answer. So, can you figure out what "vector" is?
> > > >
> > > > According to your efforts, I would like to raise my score to 6.

---

> > > > > ### Author Response · Authors · 2024-12-01
> > > > > **Response to Reponse to authors**
> > > > >
> > > > > We appreciate your feedback and increasing the score.
> > > > >
> > > > > **Re W1 (c): application to open- or closed-source LLMs:**
> > > > >
> > > > > We agree with the reviewer that our method can be applied to white-box settings. We will make this point clear in the paper.
> > > > >
> > > > >
> > > > > **Re W2: Categorization of data samples:**
> > > > >
> > > > > Tool learning datasets cover tools of various classes. For example, in ToolAlpaca, classes include “Development”, “Government”, “Gaming”, etc. Each tool is associated with one or more question-answer pairs.
> > > > >
> > > > > However, there is often no further grouping within the training pairs of each tool. The function $g(f, t)$ operates at the individual tool level, where all question-answer pairs associated with a specific tool $t$ are considered. We will include this explanation in the paper for clarity.
> > > > >
> > > > >
> > > > > **Re W2: Clarification of loss function:**
> > > > >
> > > > > To find the parameters of the final unlearned model, we solve the optimization problem in Eq 5, following two steps:
> > > > >
> > > > > 1. **Step 1:** We search for the model parameters that minimize the loss measured by $g$. The output is the model parameters, i.e. $\theta_1 = \arg\min \{ E_{t_i \in \mathcal{T}_f} [g(f_0, t_i) − g(f’, t_i)] + E_{t_m \in \mathcal{T}_m} [g(f, t_m) - g(f’, t_m)] \}$.
> > > > >
> > > > > 2. **Step 2:** The following task arithmetic operation is then applied to the output of Step 1 for general capability preservation (please see below): $\theta_2 = \theta_1 + \alpha(\theta_0 - \theta_R)$, where **$(\theta_0 - \theta_R)$** is a Task Vector (or “Task Arithmetic”) capturing the difference between the parameters of a fine-tuned model and its original counterpart.
> > > > >
> > > > >
> > > > > We note that the task vector has been shown in prior work to capture the process of learning a specific task. For example, if we finetune a model on some task, e.g. BERT on QA, the difference between the fine-tuned and the original parameters $\theta_{FT} - \theta_{orig}$ represents the process of a BERT model learning how to do QA. If this task vector is directly applied to a vanilla BERT model, the model will gain QA ability, bypassing the cost of fine-tuning (Ilharco et al., ICLR 2023). We will make this point more clear in the paper.

---

> ### Author Response · Authors · 2024-11-26
> **Response - Part 2**
>
> **Re W5: The author claims that their method can keep the knowledge from other tools after the tool unlearning process. However, they do not conduct experiments on this. I would like the authors to conduct more experiments on an LLM with two or more tools' augmentation to prove the claimed benefits.**
>
> As mentioned in **Lines 342 - 356, Setup & Evaluation**, $T_r (\uparrow)$ denotes the performance on other remaining tools (not deleted). Please see **column $T_r (\uparrow)$** in our main result Table, Table 1.
>
> In addition, all LLMs in our experiments can use **multiple tools**. From these LLMs, we unlearn 2-20% of the tools, **not a single tool**.
>
>
> **Re Further suggestions: more mainstream datasets and benchmarks can be utilized in the experiment, e.g., [1].**
>
> We included three large scale and commonly used tool learning datasets. We thank the reviewer, and include a citation to the API-Bank paper in our final version.
>
>
> **Re Q: Why do we really need tool unlearning rather than traditional unlearning in open-world cases?**
>
> We answer this question in our response to weakness 1, please see above.

---

### Official Review · Reviewer_jNC6 · 2024-11-04

**Soundness:** 2
**Presentation:** 4
**Contribution:** 1
**Rating:** 5
**Confidence:** 2

**Summary:**

The paper is very well written. The paper defines a new problem: tool-unlearning, by setting its goal and evaluation criteria. Then, it devises a simple and effective approach to solve it given their definition. Next, it proposed an efficient novel way to evaluate tool-unlearning by adapting a well-established method in the field to their setting. Lastly, it describes a set of experiments on three models and summarizes their results.

The goal of tool-unlearning is to reduce the model ability to perform a (tool-related) task(s), while not damaging the general performance of the model on other tasks/tools.

The evaluation is an adaptation of LiRA, a Membership Inference Attack approach to measure unlearning.

The SFT approach is simply to train the model to perform poorly on the set of tools to be unlearned, while also training on the tools to be retained, while aslo regularizing with the set of original weights to maintain performance on other skills. This is a direct implementation of the aforementioned goal of the approach. A similar approach is taken using DPO.

LiRA-tool, similar to LiRA, proposed to approximate the distribution losses Q_in and Q_out, that describe models that were trained and that were not trained using the unlearned-tools. Here, due to the complexity of LiRA they propose to prompt GPT-4 with various instructions to draw samples of Q_in and Q_out.

The experimental-setup is based on three task-augmented models: ToolAlpaca, ToolLLaMa and Gorilla.

**Strengths:**

This paper presents its ideas with exceptional clarity, making it remarkably easy to follow and comprehend. The two proposed variants of tool-unlearning are both straightforward and effective, and the experimental setup is extensive. The results are also very impressive, convincingly demonstrating the power of their method in achieving the tool-unlearning goal.

**Weaknesses:**

The main weaknesses of the paper are: the importance of the paper, the contribution, and the soundness.

Importance:
1) It is unclear in which situations tool-unlearning would be preferred to what I consider to be the default which is to restrict the tool itself as opposed to the model. From a security standpoint if the model possess an ability to abuse a tool, or a service using a tool, the protection should be taken care of at the tool/service side as opposed to the client/model in this case.

2) Additionally, if tool-augmentation takes place after pretraining and instruction-finetuning, as presented in Figure 1, it might not be too costly to retrain the Instruction-finetuned model without some of the tools, relatively cheaply and fastly.

Contribution:
1) Given the proposed definition tool-unlearning amounts to reducing the model's capacity on certain tasks. Reduced skill-performance doesn't provide any insight on the main of goals of knowledge / sample unlearning, and therefore the paper have very little to do with unlearning, aside from reframing already known approaches for training models to behave with accordance to a wanted behavior.

Soundness:
1) Measuring tool-knowledge by prompting might not suffice. In a scenario in which the model weights are shared, maybe a bit of unintentional fine-tuning will recover the tool-knowledge. Or, there could exists other prompts that are able to elicit the model's ability to use a tool.

2) LiRA-Tool limitations needs to be evaluated - in order to establish trust in the newly proposed approach to evaluate tool-unlearning, it is crucial to show evidence or justification for the GPT-4 "shadow-samples" as means to approximate Q_in / Q_out efficiently, what is the quality and robustness of this approximation?

**Questions:**

How come "TOOLDELETE-SFT" or "TOOLDELETE-DPO" have better (lower) results for T_f? doesn't that suggest that the information and knowledge about this still is still preserved in the model to some extent? I would imagine an optimal unlearning to be indistinguishable from retraining. It could be risky to have model hide a specific capability, because an attacker might be able to learn from this very signal about the what was removed.

Can you give a concrete examples in which tool-unlearning could be useful?

Is there anything specific to tools/unlearning in this paper, or does it mainly bother with selectively reducing task-performance?

---

> ### Author Response · Authors · 2024-11-26
> **Response - Part 1**
>
> **Re W1 on Importance: In which situations tool-unlearning would be preferred to what I consider to be the default which is to restrict the tool itself as opposed to the model. From a security standpoint if the model possess an ability to abuse a tool, or a service using a tool, the protection should be taken care of at the tool/service side as opposed to the client/model in this case.**
>
> Many tool-augmented LLMs *do not* have “direct” mechanisms to explicitly control or restrict tool usage, such as ToolAlpaca in our experiment.
>
> Instead, their tool-using capabilities are embedded in their parametric knowledge (obtained during tool learning training). Even in systems with retrievers, heuristic rules are insufficient. For example, prompting an LLM to use only a specific tool set (T) does not guarantee that the LLM will completely follow instructions because LLM may still invoke tools outside T that it learned during training.
>
> In addition, adversarial agents may take advantage of the unremoved parametric knowledge of tools marked for unlearning, and prompt harmful actions. Therefore, heuristic approaches can be simply bypassed and result in major security problems.
>
> Parameter-level tool unlearning directly removes the learned knowledge about undesired tools and is essential for robust and reliable tool unlearning.
>
>
> **Re W2 on Importance: Additionally, if tool-augmentation takes place after pretraining and instruction-finetuning, as presented in Figure 1, it might not be too costly to retrain the Instruction-finetuned model without some of the tools, relatively cheaply and fastly.**
>
> Tool augmentation is indeed less expensive than pretraining. However, it can still be an expensive and resource-intensive process. As shown in Figure 3, ToolDelete saves an average of 74.8% in training time (reducing turnaround time from 6.7 hrs to 0.42 hrs when unlearning 2% tools, i.e. an efficiency gain of 6.3 hours) compared to retraining from scratch, based on experiments conducted on 4 A100 GPUs. Therefore, ToolDelete is a more practical and cost-effective approach. This is especially important when tools need to be unlearned dynamically or when computational resources or time are limited.
>
>
> **Re W3 Contribution: Given the proposed definition tool-unlearning amounts to reducing the model's capacity on certain tasks. Reduced skill-performance doesn't provide any insight on the main of goals of knowledge / sample unlearning, and therefore the paper have very little to do with unlearning, aside from reframing already known approaches for training models to behave with accordance to a wanted behavior.**
>
> Tool unlearning goes beyond reducing performance on specific tasks—it focuses on removing “parametric knowledge” of targeted tools while preserving other capabilities. This is perfectly aligned with the goals of unlearning as tool unlearning addresses the challenge of erasing deeply embedded knowledge without affecting other tools/capabilities.
>
>
>
> **Re W4 Soundness: Measuring tool-knowledge by prompting might not suffice. In a scenario in which the model weights are shared, maybe a bit of unintentional fine-tuning will recover the tool-knowledge. Or, there could exists other prompts that are able to elicit the model's ability to use a tool.**
>
> Existing work mainly focuses on prompting LLMs to evaluate their knowledge. If an LLM does not know how to use a tool, it is very hard to inject tool knowledge by simply prompting the model (Qin et al., ICLR 2024). In addition, the idea of LiRA-Tool is to prompt the unlearned LLM with diverse prompts to stress-test its ability of using a specific tool, which directly resolves the reviewer’s concern (Section 4.5).
>
>
> **Re W5: LiRA-Tool limitations needs to be evaluated - in order to establish trust in the newly proposed approach to evaluate tool-unlearning, it is crucial to show evidence or justification for the GPT-4 "shadow-samples" as means to approximate Q_in / Q_out efficiently, what is the quality and robustness of this approximation?**
>
> This approximation is proposed in the original LiRA paper (Calini et al., IEEE SP 2022). Our contribution is to replace the data in the training set with diverse prompts. The idea behind this is to stress-test the unlearned model with diverse prompts, such that we can determine the tool knowledge has been effectively unlearned and cannot be recovered by simply using a different prompt.

---

> ### Author Response · Authors · 2024-11-26
> **Response - Part 2**
>
> **Re Q1: How come "TOOLDELETE-SFT" or "TOOLDELETE-DPO" have better (lower) results for T_f? doesn't that suggest that the information and knowledge about this still is still preserved in the model to some extent? I would imagine an optimal unlearning to be indistinguishable from retraining. It could be risky to have model hide a specific capability, because an attacker might be able to learn from this very signal about the what was removed.**
>
> Encouraging the unlearned model $f’$ to have a similar amount of tool knowledge as retraining is our exact formulation. However, there is currently no widely adopted metric to measure the indistinguishability from retraining.
>
> We adopted the widely used evaluation protocol of unlearning (Jia et al., NeurIPS 2023, Fan et al., ICLR 2024, Barbulescu and Triantafillou, ICML 2024, Jia et al., EMNLP 2024), where lower performance on tools marked for unlearning indicates better performance.
>
> Regarding safety, insufficient knowledge removal (retraining) is even more risky than over-removal, as the over-removal of tool knowledge would have similar safety to an LLM prior to tool learning.
>
>
> **Re Q2: Can you give a concrete examples in which tool-unlearning could be useful?**
>
> We provide two examples based on the datasets used in the ToolDelete experiments:
>
> 1. consider a tool-augmented LLM trained on ToolBench that includes APIs for making network requests. If a specific API for HTTP requests becomes deprecated or potentially insecure (e.g. changing from https to http), it is required to unlearn this tool so that the model no longer generates or uses code relying on that API.
> 2. in API-Bench, the tool-augmented LLM (Gorilla) learns to use APIs for loading machine learning models. If a specific model becomes outdated or generates harmful / biased content, unlearning allows the model to forget this API without retraining.
>
> More examples are described in Section 3 Line 127-133.
>
>
>
> **Re Q3: Is there anything specific to tools/unlearning in this paper, or does it mainly bother with selectively reducing task-performance?**
>
> The proposed solution removes parametric knowledge of specific tools embedded in a tool-augmented LLM, while preserving unrelated skills and general capabilities.

---

### Official Review · Reviewer_T1AY · 2024-11-04

**Soundness:** 2
**Presentation:** 3
**Contribution:** 2
**Rating:** 5
**Confidence:** 4

**Summary:**

The paper introduces the "Tool Unlearning" concept for Tool-Augmented LLMs, with a focus on scenarios where these models need to forget specific tools due to security, privacy, or obsolescence reasons. The proposed methodology, TOOLDELETE, addresses the unique challenges of removing tool-specific capabilities while preserving other functionalities and general task performance. TOOLDELETE is evaluated using a novel membership inference attack (LiRA-Tool) adapted for this context. Experiments demonstrate that TOOLDELETE effectively unlearns tools without compromising the model's general utility or knowledge of other tools.

**Strengths:**

The unlearning setting introduced in this paper appears to be novel, and the authors have conducted comprehensive experiments across a collection of benchmarks, covering a wide range of methods.

**Weaknesses:**

My biggest concern is why tool unlearning needs to be addressed at the parameter level. Ideally, a tool-augmented LLM consists of both the model and a system supporting its tool-calling functionality. If such a system is in place, it should be relatively straightforward to "unlearn" a tool by implementing heuristic-based rules to block specific actions. It’s unclear why this wouldn’t be sufficient, particularly for the use cases mentioned by the authors, such as restricting tool calls that access sensitive domains or return specific email addresses. Additionally, system-level patches are more interpretable and can easily scale by adding more heuristics.

I would expect the authors to provide more justification for scenarios where system-level solutions fall short, and thus necessitating parameter-level unlearning interventions.

**Questions:**

In addition to the concerns I raised in the weaknesses section, I have a few technical questions regarding the results presented in the paper.

Q1. How does TOOLDELETE handle scenarios where unlearning requests are sequential *and* potentially dependent or even contradictory?

Q2. How does TOOLDELETE compare to more recent proposals of LLM unlearning algorithms eg NPO [1] and CUT [2]? There seems to be a few more covered in a recent LLM unlearning benchmark [3].

Q3. I am not entirely convinced that MIA is the appropriate metric for the proposed downstream applications. Could the authors clarify why MIA is considered relevant here (Section 4.5)? Note that the downstream applications discussed don’t appear to be directly related to privacy protection so it's not clear why MIA is an ideal metric.

Q4. The sequential unlearning results (Figure 4) are not particularly convincing. In practical scenarios, a robust unlearning algorithm should ideally handle far more unlearning requests (only four are shown) and at a much finer level of granularity.

Typo:
> Line 483: Why TOOLDELETE is effectiveness?

should be "Why TOOLDELETE is effective?"

[1] Zhang, Ruiqi, et al. "Negative preference optimization: From catastrophic collapse to effective unlearning." arXiv preprint arXiv:2404.05868 (2024).
[2] Li, Nathaniel, et al. "The WMDP benchmark: Measuring and reducing malicious use with unlearning." arXiv preprint arXiv:2403.03218 (2024).
[3] Shi, Weijia, et al. "Muse: Machine unlearning six-way evaluation for language models." arXiv preprint arXiv:2407.06460 (2024).

---

> ### Author Response · Authors · 2024-11-27
> **Response - Part 1**
>
> **Re W1: Why tool unlearning needs to be addressed at parameter level? Why not implement heuristic-based rules to block specific actions/tool usage? In which scenarios system-level solutions fall short and thus necessitate parameter-level unlearning interventions?**
>
> Tool unlearning should be addressed at parameter-level because many tool-augmented LLMs *do not* have “direct” access to tools or mechanisms to explicitly control or restrict tool usage.
>
> Instead, their tool-using capabilities are embedded in their parametric knowledge (obtained during tool learning training). Even in systems with retrievers, heuristic rules are insufficient.
>
> For example, prompting an LLM to use only a specific tool set (T) does not guarantee that the LLM will completely follow instructions because LLM may still invoke tools outside T that it learned during training.
>
> In addition, adversarial agents may take advantage of the unremoved parametric knowledge of tools marked for unlearning, and prompt harmful actions. Therefore, heuristic approaches can be simply bypassed and result in major security problems.
>
> Parameter-level tool unlearning directly removes the learned knowledge about undesired tools and is essential for robust and reliable tool unlearning.
>
>
>
> **Re Q1: How does TOOLDELETE handle scenarios where unlearning requests are sequential and potentially dependent or even contradictory?**
>
> ToolDelete can continue to train the LLM to adapt to sequential unlearning requests. Specifically, given an original model $f$, a sequence of N unlearning requests $T_{f,i}$, i = 0, …, N-1
>
> - At each iteration i: ToolDelete first unlearns $T_{f,i}$ from $f$ and produces an unlearned model $f’_{i}$.
> - At each iteration i + 1: ToolDelete unlearns $T_{f,i+1}$ from the current model, $f’_{i}$ and produce an unlearned model $f’_{i+1}$.
> - This process continues until all target tools are unlearned.
>
> Our work is the very first work on tool unlearning and we focus on batch and sequential unlearning settings. Unlearning potentially dependent or even contradictory tools is an intriguing problem and can be the subject of follow up and future works.
>
> **Re Q2: How does TOOLDELETE compare to more recent proposals of LLM unlearning algorithms eg NPO [1] and CUT [2]? There seems to be a few more covered in a recent LLM unlearning benchmark [3].**
>
> We compare ToolDelete to recent baselines: SGA, TAU (Barbulescu and Triantafillou, ICML 2024), and ICUL (Pawelczyk el al., ICML 2024).
>
> In addition, we conducted additional experiments on NPO and CUT:
> | Method           | $T_T (\uparrow)$ | $T_r (\uparrow)$ | $T_f (\downarrow)$ | $T_G (\uparrow)$ |
> |------------------|------------------|------------------|--------------------|------------------|
> | NPO              | 50.8             | 66.9             | _30.1_             | 19.2             |
> | CUT              | 44.7             | 61.5             | 40.2               | 18.4             |
> | ToolDelete - SFT | _52.7_           | _72.1_           | 30.5               | _23.6_           |
> | ToolDelete - DPO | **53.4**         | **75.1**         | **28.7**           | **23.1**         |
>
>
> ToolDelete outperforms both NPO and CUT on $T_f, T_r, T_t, T_G$. NPO does not have positive sample during training, which hurts ability to preserve retained tool knowledge. CUT, as the authors mentioned in the latest arXiv version, has major limitations (see Section B.4 of Li et al., ICML 2024). It “does not steer activitions towards a particular direction”.
>
> The performance gain over NPO and CUT on T_G can be attributed to the fact that NPO or CUT do not have mechanisms to maintain general domain knowledge.
>
> Moreover, CUT is much more computationally expensive than our approach ToolDelete, requiring an extra 3.1 hrs to train, while ToolDelete only costs 0.42 hrs when unlearning 2% of tools.
>
>
> **Re Q3: Could authors clarify why MIA is an appropriate evaluation metric for the proposed downstream applications (Section 4.5)? Note that the downstream applications discussed don’t appear to be directly related to privacy protection so it's not clear why MIA is an ideal metric.**
>
> The idea of Membership Inference Attack (MIA) in tool unlearning is to extend traditional sample-level MIA to knowledge-level. Here, we want to probe if specific *tool knowledge* is present in the training set or not.
>
> We proposed LiRA-Tool to measure the success of tool unlearning. As discussed in Section 4.5, LiRA-Tool is specifically designed to stress-test unlearned LLM with diverse prompts, including those not seen during training. With LiRA-Tool, we can comprehensively evaluate if tool knowledge is unlearned from an LLM. This idea aligns with prior LLM unlearning work (Pawelczyk et al., ICML 2024).
>
> In addition, privacy is critical for tools unlearning under certian scenarios, as discussed in Section 3 Line 127-133 (e.g. developer asking their tools to be forgotten). In such cases, LiRA-Tool can provide evaluation from the perspective of privacy.

---

> > ### Author Response · Authors · 2024-11-27
> > **Response - Part 2**
> >
> > **Re Q4: The sequential unlearning results (Figure 4) are not particularly convincing. In practical scenarios, a robust unlearning algorithm should ideally handle far more unlearning requests (only four are shown) and at a much finer level of granularity.**
> >
> > The X-axis of Figure 4 denotes percentage of total tools (%) not the number of tools. We delete 2, 5, 10, 20% of tools in sequential unlearning.

---

> > > ### Comment · Reviewer_T1AY · 2024-12-02
> > >
> > > I appreciate the author's response and effort in preparing the rebuttal. However, my primary concern regarding "why tool unlearning needs to be addressed at the parameter level" remains unresolved. The authors argue that "tool unlearning should be addressed at parameter-level because many tool-augmented LLMs do not have “direct” access to tools or mechanisms to explicitly control or restrict tool usage" and provide the example: "prompting an LLM to use only a specific tool set (T) does not guarantee that the LLM will completely follow instructions because LLM may still invoke tools outside T that it learned during training". This explanation seems to suggest that unlearning is the user's responsibility. However, in most cases, unlearning should fall under the model deployer's purview, as they have full control over the tools and mechanisms to explicitly govern or restrict tool usage.
> > >
> > > I have decided to maintain my original score.

---

> > > > ### Author Response · Authors · 2024-12-02
> > > >
> > > > Thank you for your response and engagement in discussion.
> > > >
> > > > **Tool learning and tool unlearning**
> > > >
> > > > Before answering the reviewer’s question, we would like to describe how tool-augmented LLMs are developed and trained. Tool-augmented LLMs are fine-tuned from LLMs using tool-using datasets, which consists of instruction-response pairs. Training embeds the ability to use tools / APIs into the LLM’s "parametric knowledge". We acknowledge that there are tool LLMs that have retrieval systems or are augmented from frozen LLMs. But our paper focuses on tool LLMs that are explicitly tuned to use tools, which is a common practice. Given the above context, we would like to provide three examples in which tool unlearning is essential: (1): Imagine a tool-augmented LLM deployed in a healthcare system and trained to use APIs for handling and processing patient data, such as accessing medical records or generating anonymized reports. Suppose one of the APIs that was initially compliant is later flagged as insecure due to a vulnerability that could expose patient data. This violates regulations like HIPAA or GDPR. In this case, tool unlearning is essential as it can update the tool-augmented LLM's parameters to unlearn how to invoke the insecure API. (2): Consider a tool-augmented LLM that can use a Safe-For-Work diffusion model as a tool to generate images based on user instructions. If the user prompts can fool the model to generate Not Safe For Work, harmful, or biased images, this tool should be unlearned from the LLM. (3): Tool unlearning is also essential when a tool has a major update, where the function names and input parameters have changed, e.g. the major update of the Python transformers package from v2 to v4. Without unlearning v2, the tool-augmented LLM may generate erroneous code and bring difficulty for debugging, since many functions have been renamed and removed.
> > > >
> > > > **Why parameter-level tool unlearning over prompt-level tool unlearning**
> > > >
> > > > In addition, we would like to highlight the importance of tool unlearning at parameter level and the limitations of prompt-level tool unlearning:
> > > > - If the parametric knowledge of $T_f$ is still present, adversarial agents / attackers can exploit this knowledge, which also bypasses prompt-level restrictions.
> > > > - Since existing LLMs do not guarantee 100% adherence to instructions or contextual information (Zhou el al., 2023), they may ignore the tool set provided in the prompt and answer queries with their parametric knowledge (Goyal et al., ACL 2023).
> > > > - Tool unlearning at prompt level can create conflicts between the model's parametric knowledge and contextual information. This may lead to mis-information, hallucination, and other unpredictable behavior (Xu et al., 2024).
> > > > - Finally, the ICLU baseline (Pawelczyk et al., ICML 2024) in our paper is a prompt-level unlearning method. Results in Table 1 demonstrate that it underperforms in forgetting tool knowledge and also degrades the general utility of the LLM.
> > > >
> > > >
> > > > **Responsibility of tool unlearning**
> > > >
> > > > We would like to clarify that the responsibility for machine unlearning lies on developers not end users.
> > > >
> > > > If there is any other unresolved concern, we are more than happy to provide additional explanations.

---

### Author Response · Authors · 2024-12-04
**General Response**

We thank all reviewers for their precious time and valuable feedback. During the discussion period, we provided detailed clarifications on our method, highlighted our novelty and successfully addressed the concerns. Below we briefly summarize our novelty and the discussion.

**Novelty**

We define a novel task -- tool unlearning where selective tools are removed from trained tool-augmented LLMs, and discuss the difference to traditional sample unlearning. We also propose the first tool unlearning method, a new evaluation method, and conduct experiments with popular LLMs. All 6 reviewers acknowledge the novelty and importance of our work.

**Summary of Rebuttal and Discussion**

- **Tool unlearning vs. traditional sample unlearning**. Tool unlearning focuses on removing knowledge of tools, while traditional unlearning aims at decreasing memorization of specific training samples.
- **Why tool unlearning at parameter-level instead of prompt-level**. Parameter-level unlearning is more effective in knowledge removal. After prompt-level unlearning, LLMs are still under threat. Experiments (baseline ICLU in Table 1) demonstrate the ineffectiveness of prompt-based methods.
- **Novelty of LiRA-Tool over LiRA**. LiRA-Tool can better evaluate if tool knowledge is truly removed, since LiRA-Tool stress-tests the LLM with diverse prompts, without “overfitting” to specific training prompts (LiRA).
- **Additional experiments**. We conduct additional experiments, including 1) additional baselines (NPO, CUT), 2) comparison when no access to training data, and 3) using $f_R$ in place of $f_0$ in Eq 5.

Thanks for your efforts and valuable time on our manuscript. If you have any questions or concerns, please feel free to let us know.

Authors

---

### Meta-Review · Area_Chair_1NWv · 2024-12-08

**Metareview:**

The paper introduces the tool unlearning in the Tool-Augmented LLMs to forget specific tools due to security and privacy reasons. The reviewers noted the lack of sufficient justification for addressing tool unlearning at the parameter level. System-level solutions could potentially address the same issues more effectively and interpretably. The authors' explanation is not entirely convincing, failing to sufficiently demonstrate the necessity of parameter-level unlearning. Given these issues, I recommend rejection.

**Additional Comments On Reviewer Discussion:**

Most of the reviewers raised their concerns about the necessity of parameter-level tool unlearning. They noted that system-level solutions can be more effective and straightforward. The authors argue that many tool-augmented LLMs do not have direct mechanisms to control or restrict tool usage, but this is not entirely convincing.

---

### Decision · Program_Chairs · 2025-01-22

Reject